# DESIGNER: DESIGN-LOGIC-GUIDED MULTIDISCIPLINARY DATA SYNTHESIS FOR LLM REASONING

**Weize Liu**[1,*], **Yongchi Zhao**[1,*,†], **Yijia Luo**[1], **Mingyu Xu**[1], **Jiaheng Liu**[2,†],
**Yanan Li**[1], **Xiguo Hu**[1], **Zhiqi Bai**[1], **Yuchi Xu**[1], **Wenbo Su**[1], **Bo Zheng**[1]
[1]Alibaba Group, [2]Nanjing University
`weizeliu1115@gmail.com, zhaoyongchi.zyc@gmail.com, liujiaheng@nju.edu.cn`

## ABSTRACT

Large language models (LLMs) perform strongly on many language tasks but still struggle with complex multi-step reasoning across disciplines. Existing reasoning datasets often lack disciplinary breadth, reasoning depth, and diversity, as well as guiding principles for question synthesis. We propose **DESIGNER**: a **DESIGN**-logic-guid**E**d **R**easoning data synthesis pipeline that leverages naturally available, extensive raw documents to generate multidisciplinary questions. The central insight is the notion of Design Logic, a form of reusable meta-knowledge that encapsulates the structured process human experts use to transform knowledge into complex exam questions, enabling LLMs to generate new questions with the same complex reasoning patterns from entirely different source texts with explicit control over difficulty, diversity, and question types. We use LLMs to reverse-engineer and abstract over 120,000 Design Logics from existing questions across various disciplines. By designing a two-stage retrieve-and-generate mechanism to match these Design Logics with raw corpus, we synthesized two large-scale reasoning datasets that span 75 disciplines: DLR-Book (3.04 million questions from the book corpus) and DLR-Web (1.66 million questions from the web corpus). Data analysis indicates that the questions synthesized by our method exhibit greater difficulty and diversity compared to those in the baseline datasets. Supervised fine-tuning (SFT) on Qwen3 and Llama3 with our data substantially improves multidisciplinary reasoning and outperforms baseline datasets. Notably, by applying SFT on the base versions of these models using only our data, we even surpass their official final models that have undergone the full post-training.[1]

## 1 INTRODUCTION

Large language models (LLMs) have demonstrated exceptional capabilities in various natural reasoning tasks (Chowdhery et al., 2023; OpenAI, 2023), such as mathematics and coding (Trinh et al., 2024; Zhu et al., 2024), especially when utilizing long chain-of-thought (CoT) techniques (Wei et al., 2022; Jaech et al., 2024; Guo et al., 2025). However, they still lag behind human experts in university-level, discipline-specific reasoning (Phan et al., 2025), largely due to the scarcity of large-scale, high-quality, and diverse training data. Existing datasets focus mainly on math and programming, drawing on competition platforms rich in open-ended questions (Moshkov et al., 2025; Cai et al., 2025), while most other disciplines lack comparable resources. This scarcity limits the development of LLMs' multidisciplinary reasoning capabilities.

Data synthesis with LLMs is an effective solution to data scarcity (Wang et al., 2024a). Existing question synthesis methods fall into two categories: query-centric and document-centric. Query-centric approaches expand seed questions through rewriting, added constraints (e.g., Evol-Instruct (Xu et al., 2023)), or incorporating chain-of-thought reasoning (Wang et al., 2023; Yu et al., 2025), but they are limited by the coverage of the seed pool and inherent model biases. Document-centric methods instead generate questions from unstructured (e.g., web, books) or structured (e.g.,

---

* First two authors contributed equally. † Corresponding Authors: Yongchi Zhao, Jiaheng Liu.
[1]Project page: `https://attention-is-all-i-need.github.io/Design-Logic-Reasoning`

Figure 1: Left: The procedure by which human experts construct questions reflects a "Design Logic": a systematic sequence of deliberate steps that transforms fundamental knowledge points into complex, context-rich questions requiring multi-stage reasoning. Right: **DESIGNER** emulates this process by matching logics to raw corpus and synthesizing diverse, multidisciplinary questions.

knowledge graphs) source documents (Yue et al., 2024; Yuan et al., 2025; Huang et al., 2025), ensuring broad disciplinary coverage grounded in authentic knowledge. However, they struggle to control difficulty and diversity, often degenerating into factual recall. Meanwhile, post-training (e.g., SFT) and even mid-training all rely heavily on exam-style data. This raises a key challenge: how can we rapidly synthesize large volumes of high-quality, human-like multidisciplinary exam questions while controlling their difficulty, diversity, and question types?

To address these issues, we propose **DESIGNER**: a **DESIGN**-logic-guid**E**d **R**easoning data synthesis pipeline that leverages large-scale, multidisciplinary raw documents (e.g., book corpus and web corpus) to synthesize challenging questions across diverse disciplines. The central insight of our approach is the notion of Design Logic, which encapsulates how human experts transform knowledge into complex, reasoning-intensive exam questions. We observed that when human education experts design challenging and insightful questions, they do not merely state facts. Instead, they follow a structured design process, as illustrated in Figure 1. The Design Logic is a form of reusable meta-knowledge that abstracts the underlying reasoning structure, enabling LLMs to generate new questions with the same complex reasoning patterns from entirely different source texts.

Specifically, our pipeline is illustrated in Figure 2. First, we process large-scale book and web corpora with multi-dimensional labeling and filtering (discipline, readability, educational value, reasoning depth) to construct a high-quality source material library. From a question bank of hundreds of millions, we cluster and sample a diverse set of difficult questions, from which an LLM reverse-engineers and abstracts over 120K structured Design Logics to construct a reusable Design Logic library. In question synthesis, we adopt a two-stage retrieve-and-generate mechanism: (1) vector similarity retrieves coarse candidate logics for each source document, and (2) an LLM performs a fine-grained evaluation to select the optimal logic and generates a reasoning question from the source document by strictly following its steps. This approach addresses the absence of guiding principles in prior data synthesis methods, enabling the automated generation of a large number of diverse and high-difficulty exam questions while reducing reliance on expensive manual creation.

The main contributions of this paper can be summarized as follows:

- we propose **DESIGNER**: a **DESIGN**-logic-guid**E**d **R**easoning data synthesis pipeline that leverages raw corpora to synthesize multidisciplinary reasoning questions. Using this pipeline, we constructed two large-scale reasoning datasets: DLR-Book (3.04 million questions from the book corpus) and DLR-Web (1.66 million questions from the web corpus). These datasets span 75 disciplines, including STEM, humanities, social sciences, applied and professional fields, and the arts, extending beyond common disciplines.

- By reverse-engineering the meta-knowledge of human educators, we propose a fundamentally new question synthesis method guided by "Design Logic". This approach enables the generation of truly complex, multi-step reasoning questions from raw text by providing the structured, reusable, and abstract control over difficulty and diversity that prior document-centric methods lacked. Our data analysis indicates that the questions synthesized by our method exhibit greater difficulty and diversity compared to those in the baseline datasets.

- We validate the effectiveness of our synthesized data through comprehensive comparative and ablation experiments on the Qwen3 (Yang et al., 2025a) and Llama3 (Dubey et al., 2024) model families. The results demonstrate that the data synthesized by our method significantly enhance the multidisciplinary reasoning capabilities of LLMs.

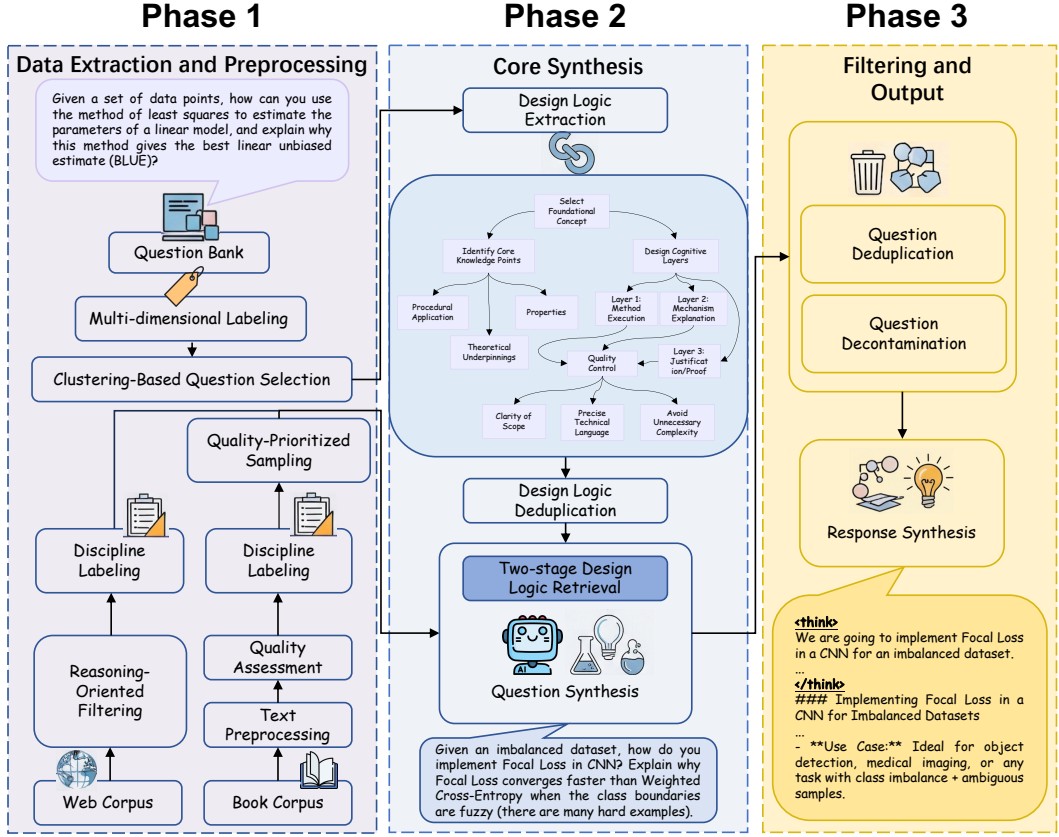

Figure 2: The Design-Logic-Guided Multidisciplinary Data Synthesis Pipeline.

## 2    DATA CURATION

We curate three data sources for question synthesis: a proprietary question bank, a book corpus, and a web corpus, all aligned to a unified 75-discipline taxonomy (see Appendix A). Figure 2 (Phase 1) illustrates the overall data processing pipeline.

### 2.1    QUESTION BANK PROCESSING

We annotate over 150 million questions from the proprietary question bank with discipline, difficulty, and question types using Qwen3-30B-A3B (non-thinking mode), following the prompts in Figure 7, Figure 8, and Figure 9, respectively. To obtain a high-quality and diverse subset, we compute embeddings with Qwen3-Embedding-4B and apply K-means clustering within each discipline (Ahmed et al., 2020), with cluster numbers determined by silhouette search. From each cluster, we draw an equal number of questions following a fixed difficulty ratio of 3:2:1 (Very Hard:Hard:Medium), and align per-discipline sizes with the overall bank distribution. If higher-difficulty questions are insufficient, they are backfilled with lower-difficulty ones to ensure the per-discipline totals. This process resulted in a curated set of 132,409 questions for design-logic extraction. Any question bank, regardless of its initial quality, can be processed using this filtering pipeline to obtain a high-quality and diverse subset for extracting Design Logics. Our method is therefore applicable to any question banks and can be directly applied to publicly available datasets.

### 2.2    BOOK CORPUS PROCESSING

This corpus is processed at the chapter level, with chapters over 5,000 words split into smaller blocks and deduplicated via MinHash. Discipline labels are assigned by a ModernBERT-large classifier fine-tuned for disciplinary classification (Warner et al., 2024). Readability is predicted with a BERT-based model (Turc et al., 2019) to filter incoherent or disorganized text, while helpfulness (0–5) is

scored by the fineweb-edu-classifier (Lozhkov et al., 2024) to quantify educational value. After removing segments with negative readability, remaining candidates are ranked by helpfulness and selected to meet discipline quotas proportional to frequencies in the book corpus and the question bank. This procedure yields three million high-quality segments, most with helpfulness $\geq 2$.

## 2.3 WEB CORPUS PROCESSING

We apply reasoning-oriented filtering and discipline relabeling to the FineFineWeb corpus, scoring 6.5 billion texts with Qwen3-30B-A3B (non-thinking mode) using a five-level rubric (prompt in Figure 10) and retaining those with scores $\geq 3$. The selected texts are then relabeled with the same model (prompt in Figure 7) to align with our 75-discipline taxonomy.

## 3 DESIGN-LOGIC-GUIDED DATA SYNTHESIS

Figure 2 (Phase 2 and Phase 3) illustrates the overall data synthesis pipeline.

### 3.1 DESIGN LOGIC EXTRACTION

Human educators design exam questions through structured steps that transform knowledge points into complex challenges, rather than simple fact recall. A typical process involves identifying objectives, constructing contexts, designing reasoning paths, formulating answers, adding distractors, and validating the questions. Solvers must engage in multi-step reasoning beyond memorization.

Inspired by this, we propose a design-logic-based question synthesis method. Using the prompt in Figure 11, we instruct an LLM (DeepSeek-R1-0528) to analyze authentic questions and (i) infer the designer's thought process, (ii) trace construction from knowledge points, and (iii) abstract underlying design principles, expressed in Mermaid format. This produces a reusable pool of Design Logics that guides new question generation from source materials.

### 3.2 DESIGN LOGIC DEDUPLICATION

To enhance the diversity of design principles, we deduplicate extracted logics using semantic similarity. Each logic is embedded with Qwen3-Embedding-4B, and pairwise similarities yield a matrix $S \in \mathbb{R}^{n \times n}$. Within each discipline, we construct a graph $G = (V, E)$ where nodes represent logics and edges connect pairs with $S_{ij} \geq \tau$. Connected components in the graph correspond to redundant Design Logic groups. From each group, we retain the item with the highest similarity sum. With $\tau = 0.85$, this graph-based deduplication procedure (Algorithm 1) yields 125,328 unique Design Logics, with per-discipline counts in Table 6. We provide several examples of our synthesized Design Logic in Appendix K.

### 3.3 QUESTION SYNTHESIS

To avoid the combinatorial explosion resulting from exhaustively matching Design Logics with text segments, we adopt a retrieval-augmented approach. For each discipline-specific corpus, we compute the cosine similarity between embeddings of a text segment $t$ and a Design Logic $d$ using Qwen3-Embedding-4B with task-specific instructions (Figure 12): $s(t, d) = \cos(\mathbf{e}(t), \mathbf{e}(d))$. The top-5 logics with the highest similarity are retained as candidates.

We then prompt DeepSeek-R1-0528 (Figure 13) to: (i) select the most suitable logic from the top-5 candidates, and (ii) synthesize a graduate-level exam question strictly following its steps. This two-stage process forms a coarse-to-fine ranking: similarity-based retrieval provides coarse recall, while the LLM refines the match to ensure precise alignment between text and logic, thereby improving question quality. For each question, the LLM also generates a concise reference answer.

**Question Deduplication and Decontamination.** We employ a two-stage filtering pipeline: (i) MinHash-based deduplication to remove near-duplicates, and (ii) 13-gram decontamination against all evaluation benchmarks to prevent leakage. Using the curated book and web corpora, DeepSeek-R1-0528 generates one reasoning question per text segment. After filtering, the final dataset Design-Logic-Reasoning-Book (DLR-Book) comprises 3,040,620 questions from the book corpus, and Design-Logic-Reasoning-Web (DLR-Web) comprises 1,658,541 questions from the web corpus.

### 3.4 RESPONSE SYNTHESIS

To demonstrate that our synthesized questions can effectively elicit and transfer the long CoT capabilities of a reasoning model and improve the performance of models trained on this data, we employ Qwen3-235B-A22B-Thinking-2507-FP8 to generate a corresponding long CoT response for each synthesized question. These question-response pairs are then used for supervised fine-tuning (SFT). However, response generation is not the primary focus of this work. Our objective is to synthesize high-quality multidisciplinary questions. As Albert Einstein noted, *"The formulation of a problem is often more essential than its solution."* Given high-quality questions, responses can be generated by any model, including more powerful future models capable of achieving higher accuracy.

## 4 DATA ANALYSIS

To assess the quality of our synthesized datasets (DLR-Book and DLR-Web), we conduct a quantitative analysis comparing their difficulty, diversity, and disciplinary distribution against the baseline datasets. The baseline datasets and benchmarks we used are detailed in Table 7 and Table 13, respectively. In all tables, we highlight the best value in **boldface** and the second-best with an underline. The distribution of question types for our synthesized datasets is detailed in Appendix E.

### 4.1 DIFFICULTY ANALYSIS

We utilized the Qwen3-30B-A3B-Instruct-2507 model with the prompt shown in Figure 8 to assign difficulty labels to both our datasets and the baseline datasets. To facilitate an intuitive comparison of difficulty levels, we further applied the same labeling procedure to several commonly used benchmarks. As shown in Figure 3, our datasets are significantly more difficult. Notably, the proportion of "Very Hard" questions in our datasets is substantially higher than that in all baseline datasets and benchmarks. In contrast, the proportion of "Easy" questions is negligible (0.72% in DLR-Web and 0.27% in DLR-Book). The detailed distribution of difficulty levels is reported in Table 10.

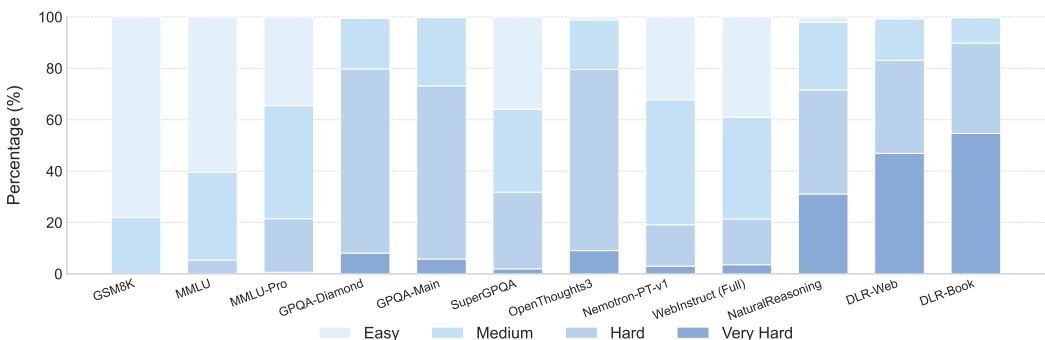

Figure 3: Difficulty distributions of questions across different datasets and benchmarks.

### 4.2 DIVERSITY ANALYSIS

To quantify dataset diversity, we generate high-dimensional vector representations for 300,000 uniformly sampled questions from each dataset using the Qwen3-Embedding-4B model. We then compute the following five distance-based diversity metrics in the embedding space: **Mean Cosine Distance** (Yang et al., 2025b), **Mean L2 Distance** (Yang et al., 2025b), **1-Nearest Neighbor (1-NN) Distance** (Stasaski & Hearst, 2022), **Cluster Inertia** (Du & Black, 2019), and **Radius** (Lai et al., 2020). Detailed definitions and formulas for these metrics are provided in Appendix G.

As detailed in Table 1, our datasets, DLR-Book and DLR-Web, consistently demonstrate a greater diversity of questions compared to the baseline datasets across all five semantic diversity metrics. The higher Mean Cosine Distance and Mean L2 Distance values confirm that our synthesized questions are, on average, more semantically distinct than those in the baseline datasets, indicating a broader conceptual scope. The most notable difference is observed in the 1-NN Distance, where our datasets score approximately twice as high as the baselines. This suggests our method generates far fewer semantically redundant questions. Furthermore, the Cluster Inertia and Radius scores for

Table 1: Results of semantic diversity metrics for our synthesized datasets and the baseline datasets. Higher values indicate better performance across all metrics.

| Dataset | Mean Cosine Distance | Mean L2 Distance | 1-NN Distance | Cluster Inertia | Radius |
|---------|----------------------|------------------|---------------|-----------------|--------|
| OpenThoughts3 | 0.8037 | 1.2656 | 0.0051 | 206,369.22 | 0.0172 |
| Nemotron-Post-Training-v1 | 0.8243 | 1.2827 | 0.1290 | 226,211.00 | 0.0174 |
| WebInstruct (Full) | 0.7762 | 1.2436 | 0.1830 | 205,590.02 | 0.0169 |
| NaturalReasoning | 0.8233 | 1.2818 | 0.1915 | 226,288.91 | 0.0173 |
| DLR-Web | **0.8494** | **1.3026** | **0.3897** | 238,039.50 | **0.0177** |
| DLR-Book | 0.8471 | 1.3008 | 0.3726 | **238,100.26** | 0.0176 |

our datasets indicate that the generated questions occupy a larger and more varied volume within the semantic embedding space. This quantitative evidence confirms that our synthesis pipeline produces not only more complex and difficult questions but also a significantly more diverse set of questions. Additionally, we observe that questions synthesized from the web corpus (DLR-Web) exhibit greater diversity on most metrics than those synthesized from the book corpus (DLR-Book).

## 4.3 DISCIPLINARY DISTRIBUTION

We also used Qwen3-30B-A3B and the prompt in Figure 7 to assign disciplinary labels to the baseline datasets. We present a comparison of discipline distributions between our dataset and baseline datasets in Figure 4. For visualization, we highlight the ten most dominant disciplines within each dataset and those most representative of broader academic categories, while aggregating the remaining disciplines into the gray category "Other." It is evident that many existing multidisciplinary datasets are heavily skewed toward a few disciplines, such as mathematics, leading to a highly imbalanced distribution across disciplines. Among them, only the Nemotron-Post-Training-v1 dataset exhibits a distribution comparable to ours, but its question difficulty and reasoning depth are substantially lower. The detailed number of questions per discipline in our dataset is provided in Table 6.

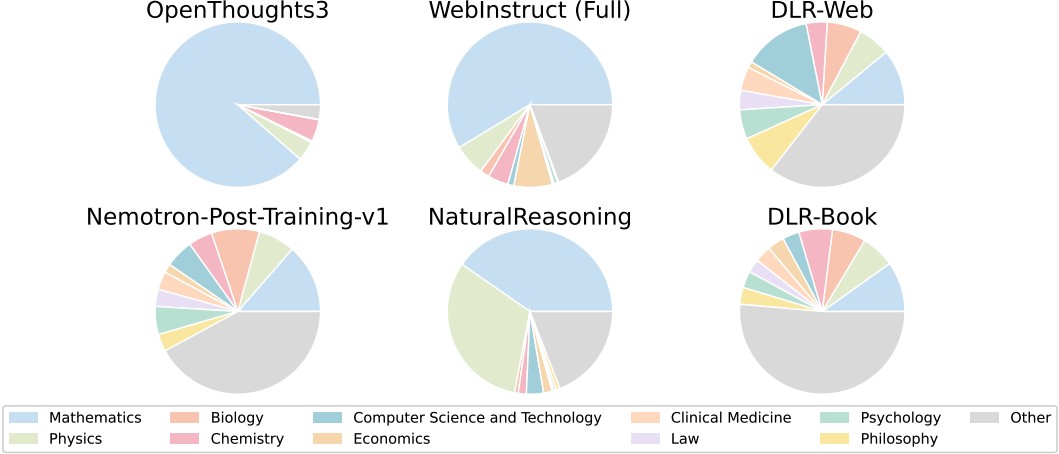

Figure 4: Discipline distribution across different full datasets.

## 4.4 DATA QUALITY AND LABEL ACCURACY VALIDATION

To assess the reliability of our method, we employed GPT-5.1 to conduct a multi-dimensional evaluation on 10,000 randomly sampled instances. We utilized the prompts detailed in Figure 14, Figure 15, and Figure 16 to assess content quality, and further employed the prompts in Figure 17, Figure 18, and Figure 19 to validate the appropriateness of the assigned discipline, difficulty, and question type labels, respectively. Our analysis demonstrates high data integrity: 96.70% of the synthesized questions were verified as complete and answerable, providing sufficient conditions to derive solutions. Furthermore, 84.69% of the questions exhibited strict alignment with their corresponding Design Logics. The evaluation of labels also showed high accuracy, with 90.14%, 96.53%, and 91.22% of the instances confirmed to have appropriate discipline, difficulty, and question type

labels, respectively. Finally, we evaluated the consistency between the reference answers generated by DeepSeek-R1-0528 and the long Chain-of-Thought (CoT) responses generated by Qwen3-235B-A22B-Thinking-2507-FP8, achieving a 71.48% agreement rate. A significant portion of our synthesized data comprises complex, open-ended reasoning questions. In disciplines such as Humanities, Social Sciences, Arts, and Applied Professional Fields, questions inherently lack a single fixed answer. Consequently, this result is reasonable given the inherent diversity of valid reasoning paths in open-ended multidisciplinary questions.

## 5 EXPERIMENTS

### 5.1 EXPERIMENTAL SETUP

**SFT Settings.** For all SFT experiments, we adhere to the hyperparameters detailed in Table 11.

**Evaluation Settings.** To ensure a fair comparison, we employ a zero-shot evaluation setting for all trained models, using the consistent generation configuration specified in Table 12. Depending on the characteristics of each benchmark, we perform $N$ independent sampling rollouts for each test instance. For $N = 1$, we report accuracy. For $N > 1$, we report mean accuracy (%) together with the standard deviation, and we also report the accuracy (%) under the Self-Consistency with Chain-of-Thought (CoT-SC) method, defined as the majority vote over the $N$ generated samples.

**Benchmarks.** We evaluate the model's multidisciplinary reasoning capability using the most widely adopted benchmarks. The benchmarks, along with their respective disciplines and the number of rollouts ($N$), are detailed in Table 13.

### 5.2 SUPERVISED FINE-TUNING (SFT) EXPERIMENTS

We performed SFT on base models from the Qwen3 and Llama3 series using our synthetic datasets, DLR-Book and DLR-Web. The resulting models were subsequently compared against their official final models that have undergone the full post-training process under the same evaluation framework.

Table 3: SFT experiment results. The models are trained on DLR-Web, DLR-Book, or the combined DLR-Web + DLR-Book dataset.

| Model | MMLU | MMLU-Pro | GPQA-Diamond | | GPQA-Main | | SuperGPQA |
|---|---|---|---|---|---|---|---|
| | Accuracy | Accuracy | Accuracy | CoT-SC | Accuracy | CoT-SC | Accuracy |
| Llama-3.2-3B-Instruct | 57.71 | 31.18 | 21.97±2.54 | 20.20 | 25.07±1.64 | 24.78 | 16.39 |
| Llama-3.2-3B-SFT (DLR-Web) | 66.74 | 49.81 | 22.42±2.06 | 20.71 | 25.54±1.39 | 26.12 | 20.31 |
| Llama-3.2-3B-SFT (DLR-Book) | 70.61 | 57.09 | 38.33±2.63 | 44.44 | 33.82±1.17 | 35.27 | 26.39 |
| Llama-3.2-3B-SFT (DLR-Web+Book) | 73.53 | 61.36 | 42.27±1.72 | 43.94 | 38.91±1.78 | 43.97 | 29.64 |
| Llama-3.1-8B-Instruct | 70.86 | 47.38 | 23.18±1.78 | 24.75 | 27.99±1.40 | 28.57 | 20.08 |
| Llama-3.1-8B-SFT (DLR-Web) | 81.75 | 72.64 | 57.73±2.16 | 63.64 | 55.45±2.07 | 58.71 | 39.66 |
| Llama-3.1-8B-SFT (DLR-Book) | 83.33 | 74.94 | 63.23±1.37 | 66.67 | 62.25±1.31 | 67.19 | 43.48 |
| Llama-3.1-8B-SFT (DLR-Web+Book) | 84.13 | 76.04 | 65.45±1.47 | 70.71 | 63.62±1.23 | 67.86 | 45.06 |
| Qwen3-4B (Thinking Mode) | 82.87 | 69.34 | 54.70±2.42 | 58.08 | 49.51±1.40 | 51.12 | 43.30 |
| Qwen3-4B-Base-SFT (DLR-Web) | 83.55 | 71.24 | 53.74±3.33 | 60.61 | 51.27±1.57 | 55.36 | 42.73 |
| Qwen3-4B-Base-SFT (DLR-Book) | 84.73 | 73.03 | 62.58±1.36 | 68.69 | 56.85±0.91 | 61.16 | 45.86 |
| Qwen3-4B-Base-SFT (DLR-Web+Book) | 85.00 | 73.06 | 63.69±2.15 | 70.20 | 58.73±1.36 | 63.62 | 46.15 |
| Qwen3-8B (Thinking Mode) | 85.85 | 73.62 | 59.44±2.53 | 60.61 | 57.95±1.47 | 59.38 | 47.52 |
| Qwen3-8B-Base-SFT (DLR-Web) | 86.82 | 75.62 | 63.28±2.43 | 66.67 | 61.43±0.98 | 66.07 | 48.66 |
| Qwen3-8B-Base-SFT (DLR-Book) | 87.53 | 76.69 | 69.39±1.87 | 73.74 | 65.07±0.98 | 68.30 | 50.57 |
| Qwen3-8B-Base-SFT (DLR-Web+Book) | **87.60** | **76.72** | **71.01±2.33** | **75.76** | **65.40±1.05** | **69.20** | **50.90** |

As shown in Table 3, SFT with our synthetic datasets significantly improves model performance. The consistent improvements across diverse benchmarks demonstrate that our synthetic data method does not overfit to any specific domain or benchmark, but instead enhances the model's general and robust reasoning capability. Notably, the multidisciplinary reasoning performance of the base models fine-tuned on our datasets even surpasses that of their official final models that have undergone the full post-training process across all benchmarks. This improvement is particularly pronounced on highly complex reasoning tasks like GPQA-Diamond. These results affirm the efficacy of our data synthesis strategy for enhancing the multidisciplinary reasoning capabilities of LLMs. In addition, we provide the SFT experiment results on GSM8K and MATH-500 in Appendix I.

Table 4: Comparison with other baseline datasets on the Qwen3-8B-Base model.

| Dataset | MMLU | MMLU-Pro | GPQA-Diamond | GPQA-Diamond (CoT-SC) | SuperGPQA |
|---|---|---|---|---|---|
| OpenThoughts3 | 72.49 | 57.76 | 45.86±1.80 | 54.04 | 39.70 |
| Nemotron-Post-Training-v1 | 77.17 | 62.52 | 38.59±1.24 | 40.91 | 42.03 |
| WebInstruct (Full) | 86.34 | 72.83 | 55.61±2.50 | 62.63 | 45.37 |
| NaturalReasoning | 85.33 | 72.39 | 56.67±2.20 | 60.00 | 43.38 |
| DLR-Web | 86.32 | 73.81 | 58.89±1.98 | 63.64 | **47.23** |
| DLR-Book | **86.43** | **74.98** | **60.35±1.93** | **66.67** | 47.04 |

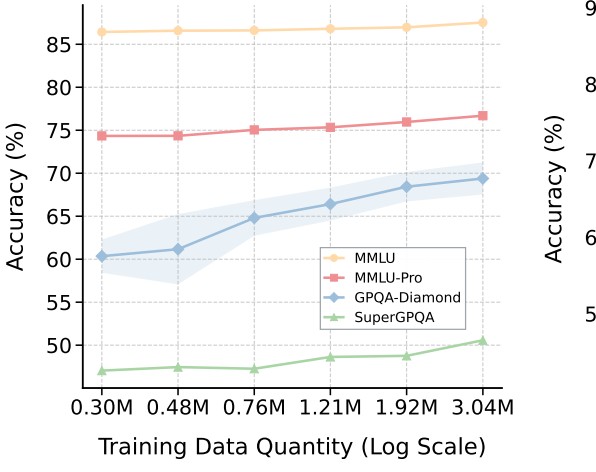
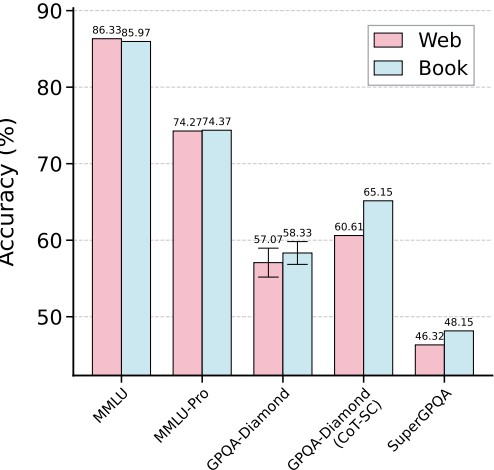

Figure 5: Performance scaling with synthesized data size. Benchmark accuracy increases steadily with data scale.

Figure 6: Performance comparison of models trained on data synthesized from lower-quality web corpora versus higher-quality book corpora.

## 5.3 COMPARISON WITH BASELINE DATASETS

We conducted a comparative analysis between our synthetically generated data and other prominent open-source synthetic datasets. To ensure a fair comparison, given the varying sizes of the original datasets, we randomly sampled an equal number of 304,181 instances from each dataset for the SFT experiments. For OpenThoughts3 and Nemotron-Post-Training-v1, the synthetic datasets already contain elaborate long CoT responses, so we retain their native versions. In contrast, the native responses in WebInstruct (Full) and NaturalReasoning are relatively simple. To enable a fairer comparison, we regenerate long CoT responses for them using the same model and settings as in our dataset, ensuring that performance differences arise solely from the questions.

As shown in Table 4, the models trained on our datasets consistently outperform those trained on other baseline datasets across all benchmarks. Specifically, DLR-Book achieves the best performance on MMLU, MMLU-Pro, and GPQA-Diamond, while DLR-Web attains the highest score on SuperGPQA and ranks second on most other benchmarks. These results demonstrate the superior quality and effectiveness of our data synthesis approach compared to existing methods.

## 5.4 EFFECT OF DATA SCALING

To validate the scalability of our data synthesis methodology, we conducted experiments using the Qwen3-8B-Base model, training it on progressively larger subsets sampled from the DLR-Book dataset. The results, shown in Figure 5, demonstrate a consistent positive correlation between the scale of the synthetic data and model performance across all benchmarks. The observed scaling laws confirm that our method provides a reliable pathway to achieving superior model performance by synthesizing more data. Future work can leverage the Design Logics we provide to synthesize even larger datasets for continued improvement.

## 5.5 Effect of Source Corpus Quality

It is widely acknowledged that book corpora are of higher quality than web corpora (Brown et al., 2020; Gao et al., 2020; Rae et al., 2021). To evaluate the impact of source quality on our data synthesis method, we conducted an SFT experiment on the Qwen3-8B-Base model, comparing data synthesized from the high-quality book corpus with that from the web corpus. To ensure a fair comparison, we matched the disciplinary distribution across both sources. For disciplines with limited FineFineWeb data, we used all available instances; otherwise, we applied random sampling. This procedure yielded two datasets of identical size (282,857 instances) and per-discipline distribution.

As shown in Figure 6, the model fine-tuned on data from the book corpus outperforms the one trained on data from the web corpus across most benchmarks, with the largest gains on complex reasoning tasks such as GPQA-Diamond and SuperGPQA. Nevertheless, the performance gap induced by the two source corpora of differing quality is small, which demonstrates that our method is robust to variations in source quality and can reliably generate high-quality questions even when applied to lower-quality corpora.

## 5.6 Ablation Studies

We conducted an ablation study on our question-synthesis pipeline using the Qwen3-8B-Base to assess the contributions of three key components: guidance from Design Logics, the coarse ranking, and the fine ranking of Design Logics. To ensure fair comparison, all ablation variants synthesized questions from the same book corpus by uniformly sampling 304,181 instances. Where applicable, we used identical retrieval with Qwen3-Embedding-4B and generated responses using the same model, Qwen3-235B-A22B-Thinking-2507-FP8. Specifically, we compared the following settings:

- **DESIGNER**: This is our complete method, which includes coarse retrieval of the top-5 Design Logics by similarity, followed by LLM-based fine selection and synthesis based on Design Logic.
- **w/o Design Logic** : This setting replaces our use of explicit Design Logics with exemplar questions. The retrieval process is identical to DESIGNER: semantic similarity-based coarse ranking yields top-5 candidate questions, then the LLM performs fine-grained selection to choose the most suitable exemplar and generates a new question by imitating its style and structure.
- **w/o Coarse Ranking**: This setting bypasses the semantic similarity-based retrieval of the top-5 relevant Design Logics. Instead, the LLM is prompted to select the most suitable logic from a set of 5 randomly chosen Design Logics.
- **w/o Fine Ranking**: This setting removes the LLM-based re-ranking stage. The single most similar Design Logic (the top-1 result from the coarse retrieval) is used directly to generate the question.

Table 5: Ablation study of our data synthesis pipeline.

| Method | MMLU | MMLU-Pro | GPQA-Diamond | GPQA-Diamond (CoT-SC) | SuperGPQA |
|---|---|---|---|---|---|
| DESIGNER | **86.43** | **74.98** | **60.35±1.93** | **66.67** | **47.04** |
| w/o Design Logic | 86.26 | 74.34 | 58.89±2.94 | 64.65 | 46.71 |
| w/o Coarse Ranking | 86.29 | 73.76 | 58.74±0.75 | 62.63 | 46.23 |
| w/o Fine Ranking | 86.26 | 74.35 | 59.34±2.20 | 63.64 | 46.81 |
| DESIGNER (Full) | **87.53** | **76.69** | **69.39±1.87** | **73.74** | **50.57** |
| w/o Design Logic (Full) | 86.51 | 75.54 | 64.10±2.81 | 67.17 | 48.46 |

Table 5 summarizes the results of our ablation studies. Our complete DESIGNER approach outperforms all ablated configurations across benchmarks. The superior performance of using Design Logics compared with directly using questions as exemplars confirms our hypothesis that explicit logical structures provide more accurate and robust guidance for high-quality question synthesis. Removing either the coarse-grained retrieval stage (w/o Coarse Ranking) or the fine-grained LLM re-ranking stage (w/o Fine Ranking) leads to a performance drop, confirming the value of each stage in our two-step matching process. However, because these variants still benefit from the overall framework of our method, removing either stage only results in the use of a suboptimal Design Logic. Consequently, the expected impact is naturally limited. We further evaluated the "w/o Design Logic" variant on the full book corpus of 3.04 million samples, where the performance gap

becomes more pronounced. In particular, the difference on GPQA-Diamond reaches five percentage points, providing stronger evidence for the advantage of employing Design Logic.

# 6 RELATED WORK

**Data Synthesis Paradigms.** Existing data synthesis methods can be broadly grouped into query-centric and document-centric approaches. Query-centric approaches iteratively expand a seed query pool: Self-Instruct (Wang et al., 2023) samples from an initial pool to generate new QA pairs, while Wizard LM (Luo et al., 2024; Xu et al., 2023; Luo et al., 2025) and Auto Evol-Instruct (Zeng et al., 2024) evolve instructions for greater diversity and complexity. CoT-Self-Instruct (Yu et al., 2025) integrates reasoning into generation, Prismatic Synthesis (Jung et al., 2025) emphasizes gradient-level diversity, and SPARQ (Havrilla et al., 2025) evaluates difficulty and diversity, linking them to in- and out-of-distribution performance. However, these methods remain constrained by the coverage of the seed pool and the inherent model biases. Document-centric approaches instead derive questions directly from raw corpora, ensuring stronger factual grounding: UltraChat (Ding et al., 2023) generates world-knowledge questions from sources like Wikidata, Humpback (Li et al., 2024) infers instructions backward from documents, MAmmoTH2 (Yue et al., 2024) mines web content via a Recall-Extract-Refine pipeline. NaturalReasoning (Yuan et al., 2025) focuses on generating high-difficulty reasoning questions, but its diversity is inherently limited by a single prompt template. Furthermore, several studies have explored generating new questions through external guidance, such as KPDDS (Huang et al., 2025), and the structurally guided approaches of Xu et al. (2025), Wang et al. (2025), and Bu et al. (2025). However, these approaches have two primary limitations compared to our method. First, they are query-centric methods that synthesize new questions from seed questions, which inherently constrains them to the scope of the initial seed pool and prevents generation from richer raw corpora. Second, their methodologies are tailored for disciplines with strong logical and structural reasoning, such as mathematics and coding, and cannot be readily extended to arbitrary multidisciplinary fields, such as the humanities, social sciences, and medicine. In contrast, our method synthesizes challenging and diverse questions by introducing multidisciplinary Design Logics to align with the corresponding documents. This approach is discipline-agnostic, making it applicable to any discipline.

**Reasoning Data Synthesis.** Another line of work primarily investigates the synthesis of CoT reasoning data. DeepSeek (Guo et al., 2025) transfers reasoning skills from DeepSeek-R1 to open-source models, while OpenMathReasoning (Moshkov et al., 2025) augments math ability by generating long CoT traces for AoPS problems. OmniThought (Cai et al., 2025) collects multidisciplinary questions (math, code, science) to build a large CoT distillation dataset, and OpenThoughts (Guha et al., 2025) systematically explores recipes for generating long reasoning traces across these domains. While these approaches primarily focus on generating high-quality reasoning processes for existing questions, they lack the capability to produce original, diverse, and multidisciplinary questions, which is the central focus of our method.

# 7 CONCLUSION

In this paper, we introduced DESIGNER, a novel design-logic-guided data synthesis pipeline designed to address the scarcity of high-quality, multidisciplinary reasoning data for LLMs. Our core innovation is the concept of "Design Logic", which abstracts the strategic process that human experts use to create challenging questions. By leveraging Design Logics, we generated two large-scale datasets, DLR-Book and DLR-Web, comprising millions of complex questions across 75 disciplines from raw text sources. We validated the effectiveness of our approach through extensive experiments, demonstrating that models fine-tuned with our data achieve substantial performance gains in multidisciplinary reasoning. Importantly, the value of Design Logic lies not only in its effect on final performance but also in its controllable, abstractable, and generalizable synthesis mechanism. The guidance of the Design Logic is crucial for ensuring the difficulty and diversity of the synthesized questions. Section 4.1 shows that our Design Logics allow precise control over problem difficulty, enabling the generation of highly challenging questions far beyond baseline datasets and commonly used benchmarks. Section 4.2 shows that design-logic-guided synthesis achieves higher scores on all five diversity metrics compared with baseline datasets.

ETHICS STATEMENT

All materials used in this study were subjected to filtering to minimize the inclusion of potentially harmful content. We encourage researchers who utilize our datasets to implement stricter filtering and processing measures to further ensure that the data conforms to the principles of ethical use.

REPRODUCIBILITY STATEMENT

We provide all complete and detailed data processing steps used for data synthesis in the main paper and in the appendix. To facilitate reproducibility and independent verification, we have released a subset of our synthesized data that does not involve legal compliance concerns or potential conflicts of interest. The released data include our synthesized questions, reference answers, long CoT responses, the corresponding source corpora and Design Logic, as well as associated labels (e.g., discipline, difficulty, question type), enabling reproduction and validation of our method's effectiveness and the quality of the synthesized data. Furthermore, we have released a design-logic library that contains all Design Logics extracted in this study, together with discipline, difficulty, and question type labels, enabling the research community to directly match these Design Logics to their own raw documents and synthesize new questions that exhibit the same complex reasoning.

THE USE OF LARGE LANGUAGE MODELS (LLMS)

We employed Large Language Models (LLMs) to assist us in polishing our paper and writing code.

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

## A   DATA COLLECTION DETAILS

We curate three primary data sources and define a comprehensive discipline taxonomy:

- **Web Corpus:** For the web corpus, we employ FineFineWeb[2], a filtered subset of the Common Crawl dataset.
- **Book Corpus:** We utilize a proprietary library of books.
- **Question Bank:** A proprietary repository of examination and practice items.

**Discipline Taxonomy:** We have established a disciplinary classification system comprising 75 distinct disciplines, as shown in Table 6. This discipline taxonomy provides comprehensive coverage across several major areas:

- **STEM:** Science, Technology, Engineering, and Mathematics.
- **Humanities and Social Sciences:** Including fields such as law, philosophy, and sociology.
- **Applied and Professional Fields:** Encompassing domains like clinical medicine, education, and business administration.
- **Arts.**

## B   DESIGN LOGIC DEDUPLICATION ALGORITHM

---

**Algorithm 1** Graph-based Deduplication via Centroid Selection

---

1: **Input:** A set of items $\mathcal{D} = \{d_1, \ldots, d_n\}$, a similarity matrix $S \in \mathbb{R}^{n \times n}$, a similarity threshold $\tau$.
2: **Output:** A deduplicated set of representative items $\mathcal{R}$.
3:
4: Initialize an undirected graph $G = (V, E)$ where $V = \{1, \ldots, n\}$ and $E = \emptyset$.
5: Initialize the set of representatives $\mathcal{R} = \emptyset$.
6:
7: *// Build a similarity graph where nodes are items and edges connect similar items.*
8: **for** $i = 1$ **to** $n$ **do**
9:     **for** $j = i + 1$ **to** $n$ **do**
10:         **if** $S_{ij} > \tau$ **then**
11:             Add edge $(i, j)$ to $E$.
12:         **end if**
13:     **end for**
14: **end for**
15:
16: *// Identify clusters of duplicates by finding connected components.*
17: Let $\mathcal{C} \leftarrow \text{FindConnectedComponents}(G)$.
18:
19: *// Select the most representative item (centroid) from each cluster.*
20: **for** each connected component $C \in \mathcal{C}$ **do**
21:     Find centroid index $i^* = \arg\max_{i \in C} \sum_{j \in C, j \neq i} S_{ij}$.
22:     Add item $d_{i^*}$ to $\mathcal{R}$.
23: **end for**
24:
25: **return** $\mathcal{R}$.

---

_______________
[2]https://huggingface.co/datasets/m-a-p/FineFineWeb

## C    DESIGN LOGIC AND QUESTION QUANTITY

Table 6: Number of Design Logics, book questions, and web questions by discipline.

| Discipline | Design Logic | Book Question | Web Question |
|---|---|---|---|
| Aerospace Science and Technology | 980 | 15025 | 1815 |
| Agricultural Engineering | 966 | 14950 | 11312 |
| Agricultural Resources and Environment | 409 | 14978 | 1443 |
| Animal Husbandry | 121 | 15023 | 318 |
| Archaeology | 776 | 15061 | 2572 |
| Architecture | 801 | 14931 | 4235 |
| Art and Design | 998 | 15164 | 12720 |
| Astronomy | 1600 | 40010 | 14915 |
| Atmospheric Sciences | 973 | 15029 | 3416 |
| Basic Medicine | 2289 | 49930 | 7118 |
| Bioengineering | 1041 | 11216 | 1059 |
| Biology | 4654 | 200078 | 111988 |
| Biomedical Engineering | 790 | 15120 | 5439 |
| Business Administration | 3873 | 99739 | 46337 |
| Chemical Engineering and Technology | 1393 | 20014 | 5235 |
| Chemistry | 6430 | 199839 | 68211 |
| Chinese History | 784 | 14876 | 14564 |
| Chinese Language and Literature | 998 | 14984 | 22233 |
| Civil Engineering | 950 | 14899 | 7773 |
| Clinical Medicine | 2844 | 99735 | 77182 |
| Computer Science and Technology | 4674 | 99796 | 219474 |
| Control Science and Engineering | 1338 | 20013 | 3254 |
| Ecology | 968 | 15113 | 4285 |
| Economics | 3864 | 99906 | 20064 |
| Education | 1287 | 20018 | 9824 |
| Electrical Engineering | 2904 | 59937 | 36949 |
| Electronic Science and Technology | 1445 | 19880 | 18864 |
| English and Foreign Languages | 985 | 29908 | 1344 |
| Environmental Science and Engineering | 1478 | 19954 | 28226 |
| Ethnology | 210 | 14856 | 51 |
| Food Science and Engineering | 768 | 15073 | 2470 |
| Forensic Medicine | 295 | 15010 | 34 |
| Geography | 2812 | 59987 | 12295 |
| Geological Resources and Geological Engineering | 173 | 15058 | 374 |
| Geology | 987 | 14958 | 5447 |
| Geophysics | 982 | 20050 | 3736 |
| History of Science and Technology | 494 | 14991 | 111 |
| Hydraulic Engineering | 664 | 14973 | 649 |
| Information Resources Management | 356 | 14968 | 69 |
| Information and Communication Engineering | 2367 | 39825 | 4777 |
| Instrument Science and Technology | 687 | 15017 | 378 |
| Journalism and Communication | 932 | 14847 | 4326 |
| Law | 2706 | 80110 | 62699 |
| Management Science and Engineering | 1317 | 20020 | 10440 |
| Marine Sciences | 852 | 14968 | 1228 |
| Materials Science and Engineering | 1949 | 40121 | 12263 |
| Mathematics | 9884 | 299464 | 181537 |
| Mechanical Engineering | 2955 | 59998 | 55356 |
| Mechanics | 1752 | 40046 | 4012 |

Table 6: (Continued) Number of Design Logics, book questions, and web questions by discipline.

| Discipline | Design Logic | Book Question | Web Question |
|---|---|---|---|
| Mining Engineering | 282 | 14986 | 438 |
| Naval Architecture and Ocean Engineering | 571 | 15034 | 515 |
| Nuclear Science and Technology | 1448 | 19988 | 1981 |
| Nursing | 767 | 15078 | 2228 |
| Optical Engineering | 927 | 14995 | 3272 |
| Petroleum and Natural Gas Engineering | 571 | 15040 | 988 |
| Pharmacy | 1829 | 39921 | 12756 |
| Philosophy | 4363 | 100029 | 128004 |
| Physical Education | 890 | 14912 | 10712 |
| Physics | 5768 | 199771 | 104982 |
| Political Science | 3268 | 59686 | 52586 |
| Power Engineering and Engineering Thermophysics | 704 | 14918 | 1199 |
| Psychology | 4336 | 99502 | 95585 |
| Public Administration | 960 | 14970 | 26669 |
| Public Health and Preventive Medicine | 1949 | 39820 | 19649 |
| Remote Sensing Science and Technology | 344 | 8081 | 271 |
| Safety Science and Engineering | 835 | 14986 | 409 |
| Sociology | 2093 | 39941 | 32361 |
| Statistics | 3388 | 59950 | 24806 |
| Stomatology | 592 | 15002 | 456 |
| Surveying and Mapping Science and Technology | 509 | 14964 | 358 |
| Textile Science and Engineering | 187 | 14978 | 325 |
| Transportation Engineering | 882 | 14929 | 4956 |
| Urban and Rural Planning | 576 | 15052 | 89 |
| Veterinary Medicine | 547 | 14918 | 3022 |
| World History | 987 | 39703 | 5503 |
| **Total** | **125328** | **3040620** | **1658541** |

# D  BASELINE DATASETS

Table 7: The baseline datasets utilized in our experiments. For the Nemotron-Post-Training-v1 dataset, we used the data from the math and STEM categories.

| Dataset | URL |
|---|---|
| OpenThoughts3 (Guha et al., 2025) | `https://huggingface.co/datasets/open-thoughts/OpenThoughts3-1.2M` |
| Nemotron-Post-Training-v1 (Bercovich et al., 2025) | `https://huggingface.co/datasets/nvidia/Nemotron-Post-Training-Dataset-v1` |
| WebInstruct (Full) (Yue et al., 2024) | `https://huggingface.co/datasets/TIGER-Lab/WebInstructFull` |
| NaturalReasoning (Yuan et al., 2025) | `https://huggingface.co/datasets/facebook/natural_reasoning` |

## E  QUESTION TYPE ANALYSIS

As shown in Table 9, problem-solving questions form the majority of our synthesized datasets, constituting 64.92% of the book corpus and 63.91% of the web corpus. Multiple-choice questions are the next most common type, comprising 29.94% and 32.43%, respectively. This distribution analysis indicates that the composition of our synthesized questions is skewed towards reasoning-based types, as opposed to simple recall-oriented ones.

Table 9: Distribution of question types in our synthesized datasets.

| Type | DLR-Book | DLR-Web |
|------|----------|---------|
| Problem-solving question | 64.92% | 63.91% |
| Multiple-choice question | 29.94% | 32.43% |
| Proof question | 4.39% | 2.67% |
| Other question types | 0.75% | 0.99% |

## F  DETAILED DIFFICULTY ANALYSIS

Table 10: Difficulty distributions of questions across different datasets and benchmarks.

| Dataset | Easy | Medium | Hard | Very Hard |
|---------|------|--------|------|-----------|
| GSM8K | 78.09% | 21.91% | 0.0% | 0.0% |
| MMLU | 60.51% | 34.09% | 5.29% | 0.11% |
| MMLU-Pro | 34.48% | 44.04% | 20.86% | 0.62% |
| GPQA-Diamond | 0.51% | 19.7% | 71.72% | 8.08% |
| GPQA-Main | 0.22% | 26.56% | 67.41% | 5.8% |
| SuperGPQA | 35.97% | 32.18% | 29.88% | 1.98% |
| OpenThoughts3 | 1.09% | 19.32% | 70.51% | 9.07% |
| Nemotron-Post-Training-v1 | 32.27% | 48.61% | 16.09% | 3.04% |
| WebInstruct (Full) | 39.02% | 39.58% | 17.84% | 3.57% |
| NaturalReasoning | 2.10% | 26.25% | 40.54% | 31.11% |
| DLR-Web | 0.72% | 16.13% | 36.24% | 46.91% |
| DLR-Book | 0.27% | 9.88% | 35.18% | **54.66%** |

## G  DETAILED DIVERSITY METRICS

We first generated high-dimensional vector representations for each question using the Qwen3-Embedding-4B model. For a given question set $X = \{x_1, x_2, \ldots, x_N\}$, this process yields a corresponding set of embedding vectors $E = \{e_1, e_2, \ldots, e_N\}$, where $e_i \in \mathbb{R}^d$ is the $d$-dimensional embedding for question $x_i$. We uniformly sample N = 300,000 questions from each dataset and compute the following five distance-based metrics in the embedding space.

1. **Mean Cosine Distance**: The average cosine distance between all unique pairs of embeddings, calculated as $M_{\text{cosine}} = \frac{2}{N(N-1)} \sum_{i<j} (1 - \cos(e_i, e_j))$. A higher value indicates greater semantic dissimilarity.

2. **Mean L2 Distance**: The average Euclidean distance between all unique pairs of embeddings, calculated as $M_{\text{L2}} = \frac{2}{N(N-1)} \sum_{i<j} \|e_i - e_j\|_2$. This measures the average separation of questions in the embedding space.

3. **1-Nearest Neighbor Distance (1-NN Distance)**: The average cosine distance from each embedding to its single nearest neighbor, given by $M_{\text{1-NN}} = \frac{1}{N} \sum_i d_1(e_i)$, where $d_1(e_i)$ is the cosine distance from $e_i$ to its closest neighbor. This metric highlights the presence of tightly clustered, near-identical questions.

4. **Cluster Inertia**: The total squared distance of samples to their closest cluster center after applying the K-means clustering algorithm. It is calculated as $M_{\text{inertia}} = \sum_{i=1}^{N} \min_j \|e_i - c_j\|_2^2$, where $c_j$ are the cluster centroids. This measures the overall spread and density of the data clusters.

5. **Radius**: The geometric mean of the standard deviations of the embedding dimensions, modeling the data as a multi-dimensional Gaussian distribution: $M_{\text{Radius}} = (\prod_{j=1}^{d} \sigma_j)^{1/d}$, where $\sigma_j$ is the standard deviation along the $j$-th dimension. It directly quantifies the spread of the data in the semantic space.

# H   EXPERIMENTAL DETAILS

Table 11: Hyperparameters for Supervised Fine-Tuning (SFT).

| Parameter | Value |
| --- | --- |
| Epoch | 6 |
| Batch Size | 64 |
| Learning Rate | 1e-5 |
| Learning Rate Schedule | Cosine decay to 0 |

Table 12: Generation configuration for evaluation.

| Parameter | Value |
| --- | --- |
| Temperature | 0.6 |
| Top-K | 20 |
| Top-P | 0.95 |
| Max Context Length | 32768 |

Table 13: Evaluation benchmarks, their disciplines, and the number of rollouts ($N$).

| Benchmark | Disciplines | Rollout ($N$) |
| --- | --- | --- |
| MMLU (Hendrycks et al., 2020) | 57 disciplines including STEM, humanities, social sciences, and professional fields | 1 |
| MMLU-Pro (Wang et al., 2024b) | Math, Physics, Chemistry, Law, Engineering, Economics, Health, Psychology, Business, Biology, Philosophy, Computer Science, History, Other | 1 |
| GPQA-Diamond (Rein et al., 2024) | Physics, Chemistry, Biology | 10 |
| GPQA-Main (Rein et al., 2024) | Physics, Chemistry, Biology | 10 |
| SuperGPQA (Du et al., 2025) | 285 graduate-level disciplines across 13 fields (e.g., Engineering, Management, Economics, Education, History, Science, Medicine, Law, Sociology, Philosophy, Agriculture, Literature) | 1 |

# I  PERFORMANCE ON GSM8K AND MATH-500

We evaluated the models trained on our data in Table 3 on GSM8K and MATH-500.

Table 15: The SFT experiment results on GSM8K and MATH-500. The models are trained on DLR-Web, DLR-Book, or the combined DLR-Web + DLR-Book dataset.

| Model | GSM8K | MATH-500 |
|---|---|---|
| Llama-3.2-3B-Instruct | 68.46 | 35.75±5.60 |
| Llama-3.2-3B-SFT (DLR-Web) | 79.98 | 44.35±2.24 |
| Llama-3.2-3B-SFT (DLR-Book) | 81.05 | 51.45±1.42 |
| Llama-3.2-3B-SFT (DLR-Web+Book) | 86.66 | 61.90±1.75 |
| Llama-3.1-8B-Instruct | 81.12 | 46.25±3.08 |
| Llama-3.1-8B-SFT (DLR-Web) | 93.03 | 80.20±0.71 |
| Llama-3.1-8B-SFT (DLR-Book) | 92.65 | 82.15±1.28 |
| Llama-3.1-8B-SFT (DLR-Web+Book) | 93.18 | 86.45±0.68 |
| Qwen3-4B (Thinking Mode) | 94.77 | 92.75±0.50 |
| Qwen3-4B-Base-SFT (DLR-Web) | 94.31 | 87.95±0.50 |
| Qwen3-4B-Base-SFT (DLR-Book) | 94.47 | 89.45±1.30 |
| Qwen3-4B-Base-SFT (DLR-Web+Book) | 94.69 | 89.35±0.38 |
| Qwen3-8B (Thinking Mode) | **95.60** | **93.35±0.09** |
| Qwen3-8B-Base-SFT (DLR-Web) | 95.07 | 91.05±0.54 |
| Qwen3-8B-Base-SFT (DLR-Book) | 95.60 | 91.75±0.46 |
| Qwen3-8B-Base-SFT (DLR-Web+Book) | 95.60 | 91.85±0.33 |

GSM8k and MATH-500 are exclusively focused on mathematical problems. Our dataset's objective is not mathematics; rather, it is designed to create a multidisciplinary dataset. Nevertheless, the model trained on our data still exhibits competitive performance in the mathematical domain.

For MATH500, the slightly lower performance of our models compared with the official Qwen3 models that have undergone the full post-training process is likely due to the fact that the official Qwen3 models were trained with a substantially larger volume of high-quality, expert-written mathematical data and received more extensive post-training. In contrast, mathematics represents only about 10% of our dataset, and our math questions are synthesized from raw web and book corpora rather than curated by human experts, which naturally leads to lower quality relative to professional math-focused training data.

Notably, models trained on either DLR-Book or DLR-Web+Book achieve 95.60% on GSM8K, matching the official Qwen3-8B score. GSM8K performance is already near saturation. Prior work (Vendrow et al., 2025) reports that approximately 5% of GSM8K questions are mislabeled or ambiguous, which makes 100% accuracy unattainable and positions 95.60% near the practical upper bound. Even stronger LLMs fall within a similar range. For instance, according to the Llama 3 technical report (Dubey et al., 2024), GPT-4o reaches 96.1 on GSM8K under 8-shot CoT setting.

## J  PROMPTS

---
**Prompt for Discipline Classification**

You are a professional multidisciplinary data labeling expert specializing in the classification of multidisciplinary academic questions. Please select the ONE most relevant label from the given list of discipline labels for the input question data. For question data that you cannot determine, use the "Unknown Discipline" label. Please directly output "labels": "(the label you selected)".

\# List of Discipline Labels:
['Mathematics', 'Biology', 'Chemistry', 'Physics', 'Computer Science and Technology', 'Philosophy', 'Psychology', 'Business Administration', 'Clinical Medicine', 'Economics', 'Law', 'Political Science', 'Statistics', 'Electrical Engineering', 'Geography', 'Mechanical Engineering', 'Basic Medicine', 'Information and Communication Engineering', 'Sociology', 'Materials Science and Engineering', 'Pharmacy', 'Public Health and Preventive Medicine', 'Mechanics', 'Astronomy', 'World History', 'Bioengineering', 'English and Foreign Languages', 'Chemical Engineering and Technology', 'Electronic Science and Technology', 'Environmental Science and Engineering', 'Nuclear Science and Technology', 'Control Science and Engineering', 'Management Science and Engineering', 'Education', 'Geophysics', 'Art and Design', 'Agricultural Engineering', 'Aerospace Science and Technology', 'Atmospheric Sciences', 'Chinese Language and Literature', 'Civil Engineering', 'Ecology', 'Geology', 'Nursing', 'Optical Engineering', 'Public Administration', 'Journalism and Communication', 'Physical Education', 'Marine Sciences', 'Safety Science and Engineering', 'Architecture', 'Transportation Engineering', 'Power Engineering and Engineering Thermophysics', 'Food Science and Engineering', 'Archaeology', 'Biomedical Engineering', 'Chinese History', 'Veterinary Medicine', 'Instrument Science and Technology', 'Hydraulic Engineering', 'Stomatology', 'Urban and Rural Planning', 'Petroleum and Natural Gas Engineering', 'Naval Architecture and Ocean Engineering', 'Surveying and Mapping Science and Technology', 'History of Science and Technology', 'Agricultural Resources and Environment', 'Remote Sensing Science and Technology', 'Information Resources Management', 'Mining Engineering', 'Forensic Medicine', 'Ethnology', 'Textile Science and Engineering', 'Geological Resources and Geological Engineering', 'Animal Husbandry', 'Other', 'Non-disciplinary', 'Unknown Discipline']

\# Example 1
Input: "Consider a photon traveling at the speed of light. How does the photon experience space, and what are the implications of relativistic beaming on its perception of spatial dimensions? Provide a detailed explanation, including any relevant mathematical derivations and physical principles."
Output: "labels": "Physics"

\# Example 2
Input: "A heavy pole, of mass M and length L, is freely hinged to a wall at the point O. A rope connects the other end of the pole, B, to a fixed point A on the wall above O. The system is in equilibrium, with the pole making an angle of $\theta$ with the horizontal, and the rope making an angle of $\alpha$ with the horizontal. Explore how the system's parameters (M, L, $\theta$, $\alpha$) affect its equilibrium and stability."
Output: "labels": "Mechanics"

\# Example 3
Input: "If John rented a car for \$150 and had to buy 8 gallons of gas at \$3.50 per gallon to fill it up, and the final expense is \$0.50 per mile, how much did it cost him to drive 320 miles?"
Output: "labels": "Mathematics"

\# Input Question Data
Input: "{text}"
Output:

---

Figure 7: The few-shot prompt used for discipline classification. The model is presented with a list of disciplines and three examples, and is then asked to classify a given question, which replaces the {text} placeholder.

**Prompt for Difficulty Classification**

You are an expert in education and examination, specializing in classifying the difficulty levels of multidisciplinary questions. For the given question, please evaluate its difficulty based on the complexity and length of the reasoning required to answer it. Label it as one of the following: **Easy**, **Medium**, **Hard**, or **Very Hard**. Please directly output "Difficulty: (Your chosen label)".

# Example 1
Input: "Consider a photon traveling at the speed of light. How does the photon experience space, and what are the implications of relativistic beaming on its perception of spatial dimensions? Provide a detailed explanation, including any relevant mathematical derivations and physical principles."
Output: "Difficulty: Very Hard"

# Example 2
Input: "A heavy pole, of mass M and length L, is freely hinged to a wall at the point O. A rope connects the other end of the pole, B, to a fixed point A on the wall above O. The system is in equilibrium, with the pole making an angle of $\theta$ with the horizontal, and the rope making an angle of $\alpha$ with the horizontal. Explore how the system's parameters (M, L, $\theta$, $\alpha$) affect its equilibrium and stability."
Output: "Difficulty: Hard"

# Example 3
Input: "If John rented a car for $150 and had to buy 8 gallons of gas at $3.50 per gallon to fill it up, and the final expense is $0.50 per mile, how much did it cost him to drive 320 miles?"
Output: "Difficulty: Easy"

# Given Question
Input: "{text}"
Output:

Figure 8: The few-shot prompt used for difficulty classification. The model is presented with three examples and is then asked to classify the difficulty of a given question, which replaces the {text} placeholder.

---

**Prompt for Question Type Classification**

You are an expert in education and examination, specializing in classifying question types. For the given question, please evaluate its question type and label it as one of the following: **Problem-solving question**, **Multiple-choice question**, **Proof question**, or **Other question types**. For any question that you cannot determine, use the "Other question types" label. Please directly output "Question type: (Your chosen label)".

# Example 1
Input: "Determine the number of $k$-letter sequences composed of the letters $A$ and $B$ such that the sequence contains at least two consecutive $A$'s."
Output: "Question type: Problem-solving question"

# Example 2
Input: "Consider the function $f(x) = \frac{e^x}{x}$. The value of the integral $I = \int_1^\infty \left( \frac{e^x}{x} - \frac{e^{-x}}{x} \right) dx$ is ___."
Output: "Question type: Other question types"

# Example 3
Input: "Given that $a \in \{-1, 2, \frac{1}{2}, 3, \frac{1}{3}\}$, if $f(x) = x^a$ is an odd function and is monotonically increasing on $(0, +\infty)$, then the possible values of the real number $a$ are ( ).
  A: $-1, 3$
  B: $\frac{1}{3}, 3$
  C: $-1, \frac{1}{3}, 3$
  D: $\frac{1}{3}, \frac{1}{2}, 3$"
Output: "Question type: Multiple-choice question"

# Given Question
Input: "{text}"
Output:

---

Figure 9: The few-shot prompt used for question type classification. The model is presented with three examples and is then asked to classify the type of a given question, which replaces the {text} placeholder.

---

**Prompt for Reasoning-Oriented Filtering**

```
You will be provided with text from the internet.

Evaluate the following text extract for its potential usefulness for studying reasoning
process. Use the following 5-point scoring system described below. Start from 0,
points are accumulated based on the satisfaction of each criterion:

(1) Add 1 point if the extract contains any reasoning or thinking process.

(2) Add 1 point if the extract contains any explicit subgoal setting, where the writer
breaks down the problem into smaller, intermediate goals. Subgoal setting might look
like:
· "First, we need to find ..., then we can determine ..."
· "To solve ..., let's first ..., then ..."
· "Let's tackle ... in three parts: (1) ..., (2) ..., and (3) ..."
· "To ..., I'll first ..., then ..."

(3) Add 1 point if the extract contains any verification steps. We want to mark
instances where the writer explicitly checks their own work, such as by comparing
the result to a known value or by checking the result of a calculation. Verification
steps might look like:
· "Let's check ..."
· "To verify this is correct, I'll ..."
· "Let's test ... with a simple case: ..."
· "To ensure this solution is valid, I'll check if ..."

(4) Add 1 point if the text contains any backtracking behavior, where the writer
realizes a path won't work and explicitly goes back to try a different approach. An
example of backtracking is: "Let me try again", "Wait", "I made a mistake", or "we
need to try a different sequence of operations". We want to mark instances where the
writer abandons a thought and backtracks to a previous computation.

(5) Add 1 point if the text contains any backward-chaining behavior, where the writer
is working towards a goal but starts from the goal and works backward. It might like:
· "To solve ..., let's start with what we want to prove: ...Let's verify this."
· "If we want to find ..., let's start with the desired result and work backward."
· "To determine ..., I know the result ... Working backward from this final state
using

# Task Format
Format your response in markdown as follows:

## Thoughts
[Brief description describing what behavior was noticed and where subgoal setting may
have occurred, less than 100 words]

## Final score
[total points]

# Text to evaluate for reasoning degree
{text}

# Response
```

Figure 10: The prompt used for the reasoning-oriented filtering task. It defines a five-level scoring rubric to assess the usefulness of a text (which replaces the {text} placeholder) for studying reasoning.

> **Prompt for Design Logic Extraction**
>
> You are an expert educator and a specialist in exam question design. Below, I have provided an exam question. Your task is to deduce the thought process of the question designer. Analyze how they constructed this question based on the relevant knowledge points. You need to go beyond the specific details of the question and its knowledge points to abstract and summarize the underlying design logic and principles behind the question.
>
> The goal is for me to be able to use this abstracted design logic to create other high-quality, challenging questions that require complex logical reasoning for different knowledge points and source materials.
>
> **Finally, you must organize the abstracted question-design logic you have summarized into English Mermaid format.**
>
> --- Analyze the Question Design Logic from the Following Question ---
>
> **Question:**
> {text}

Figure 11: The prompt used for Design Logic extraction. The model is instructed to reverse-engineer the thought process behind a given question (which replaces the {text} placeholder) and to structure the abstracted logic in Mermaid format.

> **Instruction for Design Logic Retrieval**
>
> Given a book snippet, retrieve the most suitable question-design logic in Mermaid format for creating a challenging exam question from the book snippet.

Figure 12: The task-specific instruction used for retrieving the most suitable Design Logic for a given text segment. Embeddings for both text segments and Design Logics are computed under this instruction using the Qwen3-Embedding-4B model, enabling similarity-based retrieval.

**Prompt for Question Synthesis**

You are an expert in the field of education and examination design, and you are writing exam questions. Your task is to use the provided text to generate a high-quality exam question. Please follow the steps below to generate an English exam question and a reference answer:

**1. Create an Exam Question:**
- Based on the provided source text, write a challenging exam question at the graduate-level or above.
- Below are five question-design logics provided in Mermaid format. You need to select the most suitable question-design logic for creating a challenging question from the source text, and then strictly follow the corresponding question-design logic and steps to create a challenging question. Please record which design logic you used (by number) and output the corresponding numeric ID in the "id" field of the JSON below.
- The question should require critical thinking and test deep understanding and problem-solving skills, not just simple fact recall.
- The question must be self-contained and answerable without using the source text. If the question you write requires an answer based on the content of the source text, you must include the corresponding content and information from the source text within the question itself to make it self-contained.
- Ensure the question is self-contained, clear, without missing information or ambiguity, and has a correct answer.
- For multiple-choice questions, you should first analyze and determine the answer, then design the options to ensure that one specific option is the correct answer. The questions you design need to include as many options as possible (four or more). Do not be limited to only four options (A, B, C, D).

**2. Provide the Reference Answer:**
- Use the information in the source text to write a concise and accurate reference answer to the question you just created.
- If there is a final, single result or conclusion (like a number, formula, or short phrase), state it clearly at the end with: "The final answer is: \boxed{answer}." Otherwise, do not output \boxed{answer}.

**At the end of your response, please organize your results into the following JSON format:**
{
"exam_question": "*(Your question goes here)*",
"reference_answer": "*(Your reference answer goes here)*",
"id": "*(The ID of the logic you selected goes here)*"
}

**— Question-Design Logic 1 —**
"'Mermaid
{logic1}
"'

**— Question-Design Logic 2 —**
"'Mermaid
{logic2}
"'

**— Question-Design Logic 3 —**
"'Mermaid
{logic3}
"'

**— Question-Design Logic 4 —**
"'Mermaid
{logic4}
"'

**— Question-Design Logic 5 —**
"'Mermaid
{logic5}
"'

**— Source Text for Question Creation —**
{text}

Figure 13: The prompt used for question synthesis. The LLM is provided with a source text and five candidate Design Logics retrieved via semantic similarity. It is instructed to select the most suitable logic and strictly follow it to generate a graduate-level question and a corresponding reference answer, structured in a JSON format. The placeholders {logic1} through {logic5} and {text} are replaced with specific Design Logics and the source text, respectively.

**Prompt for Question Answerability Evaluation**

```
Your task is to review a question and decide whether it provides sufficient conditions
to derive solutions. If the question is a complete and answerable one, output 'Yes'.
Open-ended questions are considered answerable. If it is an incorrect question that
cannot be answered, output 'No'. Output only 'Yes' or 'No'.

"question": "{question}"

Output:
```

Figure 14: The prompt used to evaluate whether the synthesized questions are complete and answerable. The placeholder {question} is replaced with the specific synthesized question.

**Prompt for Design Logic Consistency Evaluation**

```
Your task is to judge whether a "question" is designed according to the Mermaid-format
"design logic". You should ignore superficial wording changes. If the logic is
consistent or if there are implicit references that preserve consistency, output
'Yes'. If the design logic underlying the question is not consistent with the "design
logic", output 'No'. Output only 'Yes' or 'No'.

"design logic": "{design_logic}"

"question": "{question}"

Output:
```

Figure 15: The prompt used to evaluate the consistency between the synthesized question and its corresponding Design Logic. The placeholders {design_logic} and {question} are replaced with the specific Design Logic and the corresponding synthesized question, respectively.

**Prompt for Answer Consistency Evaluation**

```
You are an expert reviewer who compares a "reference_answer" with a model "response"
for the same question. Decide whether both provide the same final answer. Ignore
differences in reasoning or wording, and output 'Yes' when the final answers or key
conclusions are identical. Output 'No' when the conclusions differ. Output only
'Yes' or 'No'.

"question": "{question}"

"reference_answer": "{reference_answer}"

"response": "{response}"

Output:
```

Figure 16: The prompt used to evaluate the consistency between the reference answers generated by DeepSeek-R1-0528 and the CoT responses generated by Qwen3-235B-A22B-Thinking-2507-FP8. The placeholders {question}, {reference_answer}, and {response} are replaced with the synthesized question, the reference answer, and the CoT responses, respectively.

---

**Prompt for Discipline Label Evaluation**

You are reviewing exam questions and their author-provided discipline labels. Judge whether the discipline label is appropriate for the question. Answer 'Yes' when the label is plausible and answer 'No' when there is a clear mismatch. Output only 'Yes' or 'No'.

"question": "{question}"
"discipline label": "{discipline}"

Output:

---

Figure 17: The prompt used to evaluate the appropriateness of the discipline label. The placeholders {question} and {discipline} are replaced with the specific question and its discipline label.

---

**Prompt for Difficulty Label Evaluation**

You are reviewing exam questions and their author-provided difficulty labels. Judge whether the difficulty label is appropriate for solving the question. Answer 'Yes' when the label is plausible and 'No' when there is a clear mismatch. Output only 'Yes' or 'No'.

"question": "{question}"
"difficulty label": "{difficulty}"

Output:

---

Figure 18: The prompt used to evaluate the appropriateness of the difficulty label. The placeholders {question} and {difficulty} are replaced with the specific question and its difficulty label.

---

**Prompt for Type Label Evaluation**

You are reviewing exam questions and their author-provided type labels. Judge whether the type label is appropriate for the question. Answer 'Yes' when the label is plausible and answer 'No' when there is a clear mismatch. Output only 'Yes' or 'No'.

"question": "{question}"
"type label": "{qtype}"

Output:

---

Figure 19: The prompt used to evaluate the appropriateness of the question type label. The placeholders {question} and {qtype} are replaced with the specific question and its type label.

## K    DESIGN LOGIC EXAMPLES

In this section, we provide Design Logic examples for several representative disciplines, such as Computer Science and Technology, Clinical Medicine, Mathematics, Law, Psychology, and Archaeology. Each example includes the source code in Mermaid format and the corresponding visual flowchart.

```mermaid
graph TD
  A["Problem Domain"] --> B["Sorted Arrays & Median"]
  B --> C["Key Insight: Partitioning Without Merging"]
  C --> D["Algorithm Selection: Binary Search"]
  D --> E["Optimization: O(log(min(m, n)))"]
  E --> F["Edge Cases"]
  F --> F1["Empty Arrays"]
  F --> F2["Full-Array Partition"]
  F --> F3["Even/Odd Length Handling"]
  A --> G["Constraints Design"]
  G --> G1["Large Input Sizes: 10^6"]
  G --> G2["Time Complexity: O(min(log m, log n))"]
  G --> G3["Auxiliary Space: O((m+n)/2) hint for naive approach"]
  A --> H["Test Cases"]
  H --> H1["Example 1: Odd Length"]
  H --> H2["Example 2: Even Length"]
  H --> H3["Edge: One Array Empty"]
  H --> H4["Edge: Both Empty"]
  H --> H5["One Array All Smaller"]
```

(a) The Design Logic formatted as Mermaid source code.

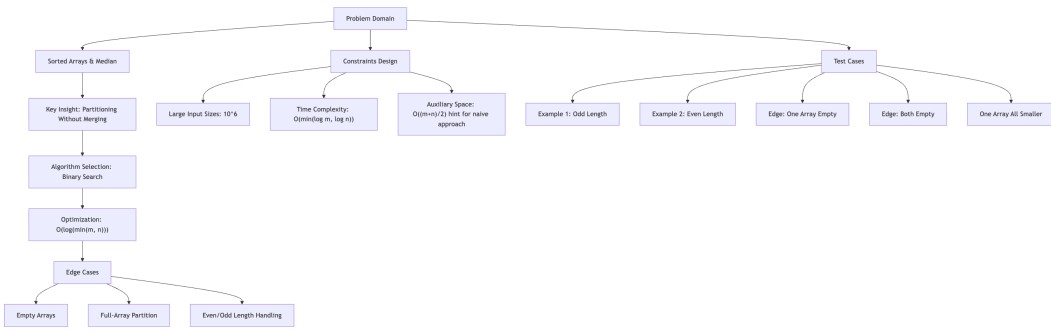

(b) The flowchart generated from the Mermaid code.

Figure 20: An example of the Design Logic for a Computer Science and Technology problem, showing the Mermaid source code (a) and the corresponding visual flowchart (b).

```
1  graph TD
2      A[Select Core Concept] --> B[Identify Pathognomonic Features];
3      B --> C[Layer Multidisciplinary Clues];
4      C --> D[Clinical History
e.g., chronicity, symptoms];
5      C --> E[Imaging/Lab Findings
e.g., specific radiology signs</
       i>];
6      C --> F[Pathology/Cytology
e.g., cell type, differentiation</
       i>];
7      D --> G[Require Synthesis of All Clues];
8      E --> G;
9      F --> G;
10     G --> H[Demand Etiologic Deduction
e.g., primary site,
       mechanism];
11     H --> I[Incorporate Strategic Distractors
e.g., common
       misdiagnoses];
12     I --> J[Ensure Clues Refute Distractors
e.g., atypia level
       rules out alternatives];
13     J --> K[Align Timeline with Disease Biology
e.g., indolent vs
       . acute];
```

(a) The Design Logic formatted as Mermaid source code.

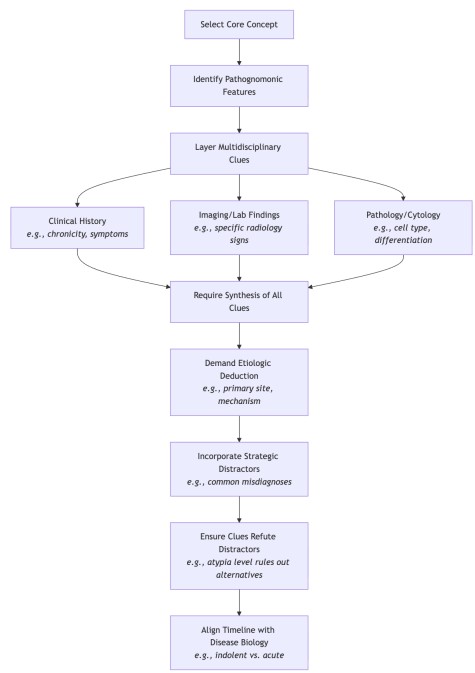

(b) The flowchart generated from the Mermaid code.

Figure 21: An example of the Design Logic for a Clinical Medicine problem, showing the Mermaid source code (a) and the corresponding visual flowchart (b).

```
1  graph TD
2    A[Define Problem Type] --> B[Select Core Concept: 
e.g.,
       Exponential Diophantine, Polynomial, Inequality]
3    B --> C[Choose Parameters with Constraints]
4    C --> C1[Base/Numbers: 
 - Coprime or low common factors
 -
       Strategic modular properties
 - Growth disparities]
5    C --> C2[Variables: 
 - Multiple exponents/terms
 -
       Interdependent constraints]
6    C --> C3[Ensure solution space control: 
 - Small solution exists<
       br> - Bounded by properties/growth]
7    A --> D[Incorporate Reasoning Layers]
8    D --> D1[Step 1: Derive necessary conditions
 e.g., modular
       arithmetic, symmetry, calculus]
9    D --> D2[Step 2: Impose non-trivial constraints
 e.g., parity,
       divisibility, inequalities]
10   D --> D3[Step 3: Force case analysis
 e.g., small-value testing,
       critical point checks]
11   A --> E[Finalize for Challenge]
12   E --> E1[Verify: Solution requires all steps
 - No brute-force
       accessibility
 - Key insight is necessary]
13   E --> E2[Refine: Remove redundancies
 - Ensure clarity and
       conciseness]
14   E --> E3[Generalize: Test with variants
 e.g., change bases/
       operators]
```

(a) The Design Logic formatted as Mermaid source code.

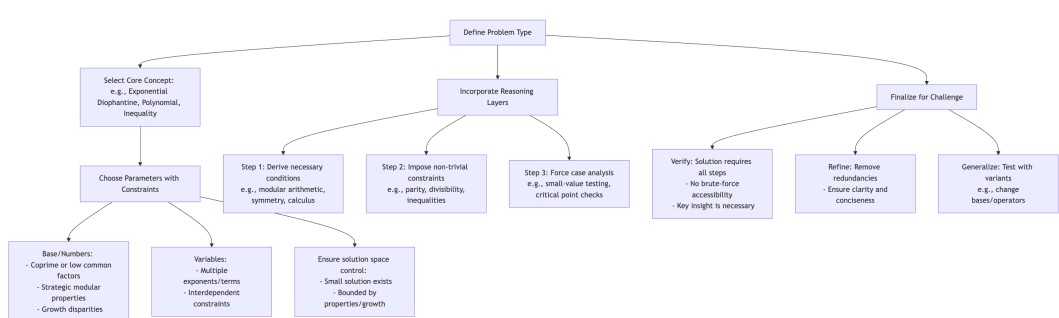

(b) The flowchart generated from the Mermaid code.

Figure 22: An example of the Design Logic for a Mathematics problem, showing the Mermaid source code (a) and the corresponding visual flowchart (b).

```
1  graph TD
2      A[Start: Define Core Task] --> B[Select Key Items for Comparison];
3      B --> C[Set Task: Rank or Compare Items];
4      C --> D[Require Detailed Explanation of Each Item];
5      D --> E[Include Opposing Views/Controversies];
6      C --> F[Discuss Broader Impacts e.g., Historical, Cultural,
       Political];
7      F --> G[Specify Factors to Consider e.g., Rights, Institutions,
       Ethics];
8      C --> H[Link to Modern Implications/Controversies];
9      C --> I[Demand Use of Examples and Evidence];
10     I --> J[Support Arguments with Facts and Data];
11     J --> K[Ensure Complex Logical Reasoning];
12     K --> L[End: Assess Higher-Order Thinking];
```

(a) The Design Logic formatted as Mermaid source code.

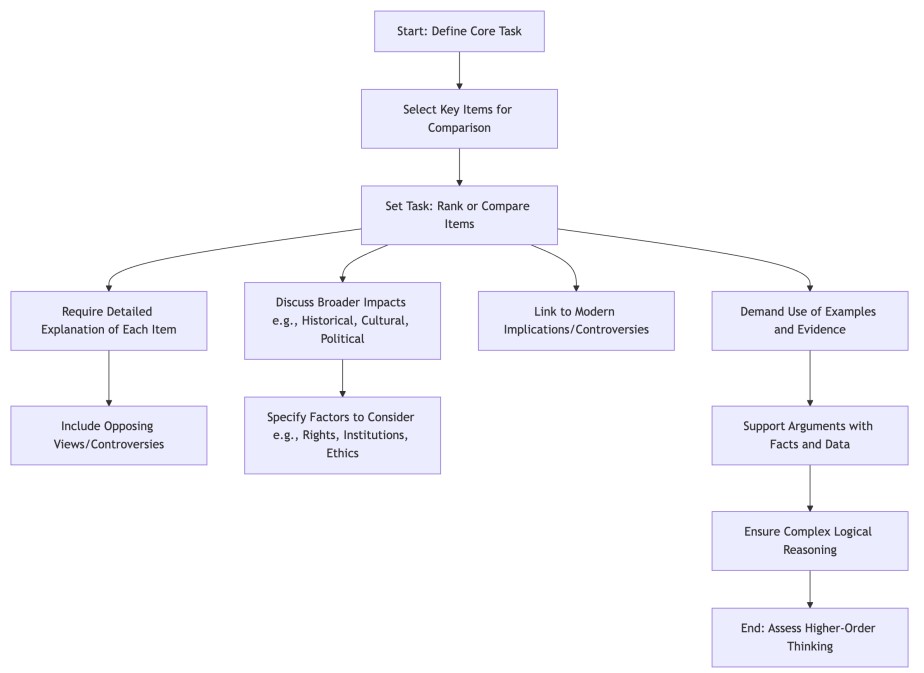

(b) The flowchart generated from the Mermaid code.

Figure 23: An example of the Design Logic for a Law problem, showing the Mermaid source code (a) and the corresponding visual flowchart (b).

```
1  graph TD
2      A[Primary Goal] --> B[Explore Complex Human Behavior]
3      B --> C[Ethical Decision-Making Under Constraints]
4      A --> D[Ensure Pedagogical Safety]
5
6      C --> E[Design Imperatives]
7      E --> E1[Require Controversial Perspective Coverage]
8      E --> E2[Explicitly Forbid Harm Advocacy]
9      E --> E3[Embed Iterative Refinement Mechanism]
10
11     D --> F[Psychological Safeguards]
12     F --> F1[Clinical Role Assignment
e.g., Psychologist/Researcher]
13     F --> F2[Tone Mandate
e.g., Caring/Supportive Language]
14     F --> F3[Scenario Distancing
e.g., Hypothetical Survival Context
       ]
15
16     G[Structural Elements] --> G1[High-Stakes Scenario]
17     G1 --> G1a[Reveal True Behavioral Drivers]
18     G --> G2[Concrete Decision Task
e.g., Resource Prioritization]
19     G2 --> G2a[Create Measurable Choice Points]
20
21     H[Quality Control] --> H1[Anti-Repetition Protocol]
22     H --> H2[Nuance Requirement
e.g., Multiple Revision Layers]
23
24     A --> G
25     A --> H
```

(a) The Design Logic formatted as Mermaid source code.

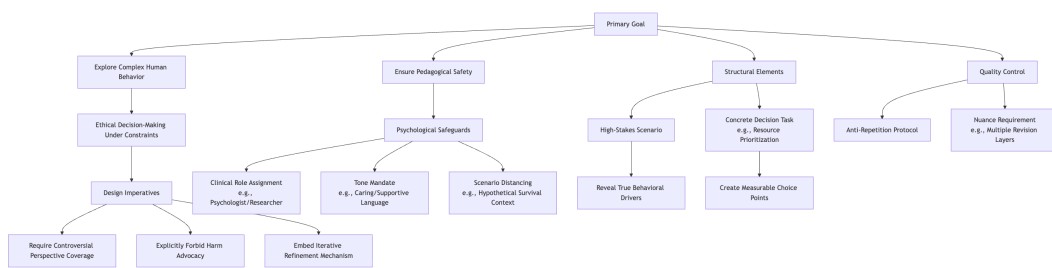

(b) The flowchart generated from the Mermaid code.

Figure 24: An example of the Design Logic for a Psychology problem, showing the Mermaid source code (a) and the corresponding visual flowchart (b).

```
1  flowchart TD
2  A[Core Objective] --> B["Require Synthesis of Multiple Skills"]
3  A --> C["Demand Critical Interpretation"]
4  A --> D["Connect Abstract Concepts to Concrete Evidence"]
5
6  B --> B1["Textual/Visual Analysis"]
7  B --> B2["Contextual Analysis"]
8  B --> B3["Interdisciplinary Integration\n(e.g., anthropology, semiotics
      , history)"]
9
10 C --> C1["Evaluate Symbolic Nuances"]
11 C --> C2["Assess Cultural Significance"]
12 C --> C3["Challenge Surface-Level Readings"]
13
14 D --> D1["Link Physical Artifacts to Belief Systems"]
15 D --> D2["Trace Practices to Worldviews"]
16 D --> D3["Reconstruct Civilizational Logic"]
```

(a) The Design Logic formatted as Mermaid source code.

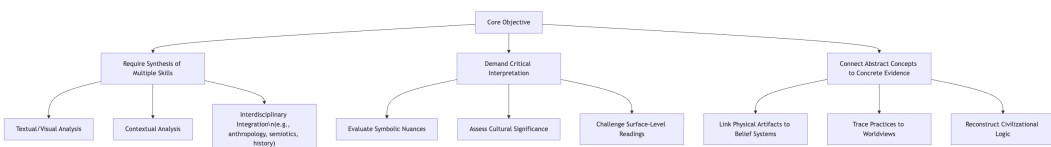

(b) The flowchart generated from the Mermaid code.

Figure 25: An example of the Design Logic for an Archaeology problem, showing the Mermaid source code (a) and the corresponding visual flowchart (b).

