# OpenReview forum: "DESIGNER: Design-Logic-Guided Multidisciplinary Data Synthesis for LLM Reasoning"
_ICLR.cc/2026/Conference — ICLR 2026 Poster_

### Official Review · Reviewer_wEhN · 2025-10-16

**Soundness:** 3
**Presentation:** 3
**Contribution:** 3
**Rating:** 6
**Confidence:** 4

**Summary:**

Large language models have achieved strong performance on many natural language processing tasks, but they struggle with complex reasoning tasks. Many existing studies use synthetic data to train large language models to improve their reasoning ability. However, these studies often lack disciplinary guidance.

To address this issue, this paper uses some unlabeled documents to generate synthetic data, including books and web data. Specifically, the process is divided into three parts: Data Extraction and Preprocessing, Code Synthesis, and Filtering and Output. The authors conduct an in-depth analysis of the synthetic data, examining data difficulty, diversity, and disciplinary distribution.

Finally, they train the model using the synthetic data and find that it leads to better training results.

**Strengths:**

+ This paper proposes a new method for data synthesis that can generate useful data for the model from unlabeled documents.

+ This paper conducts an in-depth analysis of the data, including data difficulty, data diversity, and disciplinary distribution, helping readers better understand the synthetic data and providing more information for future research.

+ The synthetic data in this paper can effectively improve the model's reasoning ability and proves to be effective across tests in multiple disciplines.

**Weaknesses:**

+ Previous studies have also generated training data from unlabeled data (such as DeepSeek-Math, JiuZhang3.0, and MAmmoTH 2.0). Even without using complex logic extraction and logic retrieval processes, their results are more significant than existing methods.

+ The process of generating synthetic data uses very large models, which incurs high costs. Does this indicate that the method has limitations in practical applications?

**Questions:**

+ The paper tests multiple-choice tasks. Does it also perform well on open-ended questions, such as GSM8k or MATH500?

+ Why does combining web data and book data lead to better results? What are the differences and characteristics of these two types of data when merged?

---

> ### Author Response · Authors · 2025-11-24
> **Response to Reviewer wEhN (1/2)**
>
> We appreciate that you find our synthesis method effective for deriving training data from unlabeled documents, our analyses of difficulty, diversity, and disciplinary coverage informative, and our synthesized data broadly beneficial for enhancing model reasoning across multiple domains. We also sincerely appreciate your recognition of our paper and the insightful comments you have provided. We have carefully considered your suggestions and further refined our work.
>
> ---
> **For Weakness 1**
>
> >Weakness 1: Previous studies have also generated training data from unlabeled data (such as DeepSeek-Math, JiuZhang3.0, and MAmmoTH 2.0). Even without using complex logic extraction and logic retrieval processes, their results are more significant than existing methods.
>
> 1. DeepSeek-Math and JiuZhang3.0 focus exclusively on the mathematics domain, representing domain-specific improvements. Our objective is to address general multidisciplinary reasoning, covering 75 disciplines.
> 2. MAmmoTH 2.0 adopts a Recall-Extract-Refine pipeline, but it lacks explicit and structured control over reasoning structure and difficulty distribution, and it often degenerates into simple question-answer pair extraction. The native responses in WebInstruct (Full) are relatively simple. To enable a fairer comparison, we even regenerate long CoT responses for them using the same model (Qwen3-235B-A22B-Thinking-2507-FP8) and settings as in our dataset, ensuring that performance differences arise solely from the questions. The WebInstruct (Full) baseline in our paper is the full dataset proposed in the MAmmoTH 2.0 paper. Through the experiments presented in our paper, we have demonstrated that the data synthesized by our method comprehensively outperforms MAmmoTH 2.0 in terms of difficulty, diversity, subject distribution, and enhancement of model performance.
> 3. Existing methods typically fall into two categories: query-centric (constrained by the coverage of the seed pool and the inherent model biases) or doc-centric (lacking control over difficulty). By reverse-engineering the meta-knowledge of human educators, we propose a fundamentally new question synthesis method guided by "design logic''. This approach enables the generation of truly complex, multi-step reasoning questions from raw text by providing the structured, reusable, and abstract control over difficulty and diversity that prior doc-centric methods lacked, which often restricted them to simple factual recall. We are the first to introduce reusable "design logic" as structured meta-knowledge to guide large-scale cross-domain synthesis. The guidance of the design logic is crucial for ensuring the difficulty and diversity of the synthesized questions.
>
> ---
>
> **For Weakness 2**
>
> >Weakness 2: The process of generating synthetic data uses very large models, which incurs high costs. Does this indicate that the method has limitations in practical applications?
>
> Our proposed methodology is compatible with LLMs of any scale. However, to ensure the synthesis of ultra-challenging questions and to maximize data quality, we utilized the most powerful available LLMs, as objective is to contribute a dataset of the highest quality to the community. We will open-source the synthesized data. By incurring the high cost of a single synthesis process, we aim to provide the community with a reusable library of design logics and a high-quality training dataset. Due to the file size limitation of the supplementary material, we have uploaded our extracted complete design logic library and a randomly sampled set of 3,000 instances from our synthesized complete dataset as an example of the dataset that we will release. These samples contain fields such as questions, long CoT responses, and labels for discipline, difficulty, and question type.

---

> ### Author Response · Authors · 2025-11-24
> **Response to Reviewer wEhN (2/2)**
>
> **For Question 1**
>
> >Question 1: The paper tests multiple-choice tasks. Does it also perform well on open-ended questions, such as GSM8k or MATH500?
>
> We evaluated the models trained on our data in Table 3 on GSM8K and MATH-500. The results are reported in the table below.
>
> | Model                                   | GSM8K | MATH-500        |
> |-----------------------------------------|-------|------------------|
> | Llama-3.2-3B-Instruct                   | 68.46 | 35.75±5.60       |
> | Llama-3.2-3B-SFT (DLR-Web)              | 79.98 | 44.35±2.24       |
> | Llama-3.2-3B-SFT (DLR-Book)             | 81.05 | 51.45±1.42       |
> | Llama-3.2-3B-SFT (DLR-Web+Book)         | 86.66 | 61.90±1.75       |
> | Llama-3.1-8B-Instruct                   | 81.12 | 46.25±3.08       |
> | Llama-3.1-8B-SFT (DLR-Web)              | 93.03 | 80.20±0.71       |
> | Llama-3.1-8B-SFT (DLR-Book)             | 92.65 | 82.15±1.28       |
> | Llama-3.1-8B-SFT (DLR-Web+Book)         | 93.18 | 86.45±0.68       |
> | Qwen3-4B (Thinking Mode)                | 94.77 | 92.75±0.50       |
> | Qwen3-4B-Base-SFT (DLR-Web)             | 94.31 | 87.95±0.50       |
> | Qwen3-4B-Base-SFT (DLR-Book)            | 94.47 | 89.45±1.30       |
> | Qwen3-4B-Base-SFT (DLR-Web+Book)        | 94.69 | 89.35±0.38       |
> | Qwen3-8B (Thinking Mode)                | **95.60** | **93.35±0.09**       |
> | Qwen3-8B-Base-SFT (DLR-Web)             | 95.07 | 91.05±0.54       |
> | Qwen3-8B-Base-SFT (DLR-Book)            | **95.60** | 91.75±0.46       |
> | Qwen3-8B-Base-SFT (DLR-Web+Book)        | **95.60** | 91.85±0.33       |
>
> GSM8k and MATH-500 are exclusively focused on mathematical problems. As stated in our response to Weakness 1, our dataset's objective is not mathematics; rather, it is designed to create a **multidisciplinary dataset**. Nevertheless, the model trained on our data still exhibits competitive performance in the mathematical domain.
>
> For MATH500, the slightly lower performance of our models compared with the official Qwen3 models that have undergone the full post-training process is likely due to the fact that the official Qwen3 models were trained with a substantially larger volume of high-quality, expert-written mathematical data and received more extensive post-training. In contrast, mathematics represents only about 10% of our dataset, and our math questions are synthesized from raw web and book corpora rather than curated by human experts, so they are inherently inferior to professional math-focused training data.
>
> Notably, models trained on either DLR-Book or DLR-Web+Book achieve 95.60% on GSM8K, matching the official Qwen3-8B score. GSM8K performance is already near saturation. Prior work [1] reports that approximately 5% of GSM8K questions are mislabeled or ambiguous, which makes 100% accuracy unattainable and positions 95.60% near the practical upper bound. Even stronger LLMs fall within a similar range. For instance, according to the Llama 3 technical report [2], GPT-4o reaches 96.1% on GSM8K under 8-shot CoT setting.
>
> ---
>
> **References**
>
> [1] Vendrow, J., Vendrow, E., Beery, S., & Madry, A. (2025). Do Large Language Model Benchmarks Test Reliability? arXiv preprint arXiv:2502.03461.
>
> [2] Dubey, A., Jauhri, A., Pandey, A., Kadian, A., Al-Dahle, A., Letman, A., ... & Ganapathy, R. (2024). The Llama 3 Herd of Models. arXiv preprint arXiv:2407.21783.
>
> ---
>
> **For Question 2**
>
> >Question 2: Why does combining web data and book data lead to better results? What are the differences and characteristics of these two types of data when merged?
>
> The performance improvement observed when training with the DLR-Book and DLR-Web datasets can be attributed to two primary factors. First, combining the two sources increases the overall training volume. Second, because DLR-Book and DLR-Web originate from distinct raw corpora, they introduce complementary knowledge and question types that enhance data diversity. The book corpus provides higher knowledge density, whereas the web corpus contributes broader topical coverage and content variety.
>
> ---
>
> Finally, we would like to express our gratitude once again for the insightful suggestions you have provided to help improve our paper. We have incorporated these suggestions in the revised paper.

---

> > ### Comment · Reviewer_wEhN · 2025-11-27
> >
> > Thanks for the detailed response. My concerns are addressed, and I will keep my score positive.

---

> > > ### Author Response · Authors · 2025-11-27
> > > **Thank you for your positive feedback!**
> > >
> > > Dear Reviewer wEhN,
> > >
> > > Thank you for acknowledging the effort we put into the rebuttal and for carefully considering our responses. We are encouraged to hear that our detailed response and the revised manuscript have addressed your concerns. Your support means a lot to us.
> > >
> > > Best regards,
> > >
> > > The Authors

---

### Official Review · Reviewer_VTR2 · 2025-10-29

**Soundness:** 3
**Presentation:** 3
**Contribution:** 2
**Rating:** 4
**Confidence:** 3

**Summary:**

This work presents a data synthesis approach aimed at alleviating the data scarcity problem in cross-disciplinary reasoning tasks. The authors construct two large-scale reasoning datasets: DLR-Book and DLR-Web, covering 75 scientific disciplines. The core contribution of this work lies in its logic-guided data design, which enhances data quality through difficulty and diversity control. Experimental results show that the proposed datasets outperform existing ones on multiple benchmarks.

**Strengths:**

1. The motivation of the work is clear and easy to understand.
2. The main contribution lies in building a large-scale multidisciplinary dataset, which achieves competitive or even superior results across multiple benchmarks (e.g., Table 1).
3. The carefully designed synthesis pipeline demonstrates thoughtful engineering and could provide useful insights for other researchers working on data generation and reasoning dataset construction.

**Weaknesses:**

1. The current approach is essentially a heuristic replacement for existing data synthesis methods, mainly relying on hand-crafted design and rejection sampling to obtain high-quality data.
2. The complexity of the data construction process might lead to error accumulation and make it difficult to control noise in the final dataset.
3. Some experimental results require further clarification:
+ In Figure 5, why does the model show better scalability only on GPQA-Diamond, while other datasets do not exhibit similar improvements?
+ In Table 5, the ablation study shows that removing Design Logic (w/o Design Logic) results in only minor performance drops. Other ablation variants also show limited improvements?
4. The data efficiency of the synthesis process is unclear. For example, during data filtering, what proportion of data was discarded? Understanding this would help assess the trade-off between data quality and synthesis cost.

**Questions:**

See Weakness

---

> ### Author Response · Authors · 2025-11-24
> **Response to Reviewer VTR2 (1/4)**
>
> Thank you very much for your detailed suggestions. We appreciate that you find the motivation of our work clear, our large-scale multidisciplinary dataset effective across benchmarks, and our synthesis pipeline thoughtfully engineered to provide useful insights. We have carefully considered your comments, and we hope that our responses below will address your concerns.
>
> ---
> **For Weakness 1**
>
> >Weakness 1: The current approach is essentially a heuristic replacement for existing data synthesis methods, mainly relying on hand-crafted design and rejection sampling to obtain high-quality data.
>
> 1. We have minimal reliance on hand-crafted design; all processes are completed via LLMs. The only truly "manual" component consists of a few prompts for synthetic data generation and labeling, which are necessary prerequisites for the LLMs to synthesize the data.
> 2. Our approach does not rely on rejection sampling. The core procedure involves a single generation pass (one question per text segment), followed only by standard data filtering, such as deduplication and decontamination. There is no extensive repeated re-sampling or heuristic discarding. The data filtering step constitutes a fundamental quality control process in data synthesis and is not the core innovation of our methodology.
> 3. Our core innovation is the introduction of a design-logic-guided reasoning data synthesis pipeline. Existing methods typically fall into two categories: query-centric (constrained by the coverage of the seed pool and the inherent model biases) or doc-centric (lacking control over difficulty). By reverse-engineering the meta-knowledge of human educators, we propose a fundamentally new question synthesis method guided by "design logic''. This approach enables the generation of truly complex, multi-step reasoning questions from raw text by providing the structured, reusable, and abstract control over difficulty and diversity that prior doc-centric methods lacked, which often restricted them to simple factual recall. We are the first to introduce reusable "design logic" as structured meta-knowledge to guide large-scale cross-domain synthesis.
> 4. Moreover, our paper provides a detailed illustration of a real data-engineering pipeline that can synthesize data capable of substantially improving LLMs' reasoning ability. This detailed pipeline is precisely what is often absent from the academic research community. Even open-source models like Qwen generally do not disclose the precise steps for their training data preparation, and such information is certainly not available for closed-source models such as GPT and Gemini.
>
> ---
>
> **For Weakness 2**
>
> >Weakness 2: The complexity of the data construction process might lead to error accumulation and make it difficult to control noise in the final dataset.
>
> Our pipeline is not a simple linear pass but a funnel model that incorporates multiple filtering and re-verification mechanisms, and the resulting SFT performance directly demonstrates that the data quality is high.
>
> 1.  We implement stringent data filtering on the raw materials at every step. The objective of these complex steps is precisely to mitigate error accumulation and enhance data quality. The final synthesized questions utilize only the extracted design logic and the filtered raw documents, making the overall synthesis not overly complicated.
>
> 2.  For matching the design logics with the filtered raw documents, we implement a two-stage retrieval process:
>     * (i) For each raw document, we use Qwen3-Embedding-4B to perform a coarse ranking of all design logics by computing semantic similarity, identifying the top-5 candidates for the given source text.
>     * (ii) We then prompt DeepSeek-R1-0528 to select the most suitable logic from the top-5 candidates. This process ensures precise matching and reduces error accumulation.
>
> 3.  Potential errors in the early steps (such as subject classification) primarily affect metadata distributions and do not compromise the content quality (logical correctness) of the final generated questions and responses. Even if the subject is mislabeled, as long as the Design Logic and the source text are correctly matched, the generated question remains high-quality data for reasoning training.
>
> 4.  The ultimate criterion for data quality is model performance. If a substantial amount of noise or logical errors were accumulated in the data, the model would typically exhibit increased hallucination or degraded performance after SFT. However, models trained on our data achieve significantly larger gains on highly complex, human-expert-level benchmarks like GPQA-Diamond, even surpassing their official final models that have undergone the full post-training process. The data scaling curve in Figure 5 indicates a continuous and stable improvement in model performance as the data volume increases. This directly demonstrates that the data contains effective reasoning signals, not accumulated noise.

---

> ### Author Response · Authors · 2025-11-24
> **Response to Reviewer VTR2 (2/4)**
>
> **For Weakness 3.1**
> >Weakness 3: Some experimental results require further clarification:
> >
> >Weakness 3.1: In Figure 5, why does the model show better scalability only on GPQA-Diamond, while other datasets do not exhibit similar improvements?
>
> - GPQA-Diamond is recognized as one of the most challenging benchmarks, specifically requiring deeper, multi-step synthesis. Our logic-guided data directly addresses this requirement, resulting in a more pronounced performance gain on GPQA-Diamond. However, performance improvements were also observed on other datasets. We also observe substantial improvements on MMLU-Pro and SuperGPQA, whereas performance on MMLU is already near saturation.
> - As shown in Table 3, the performance gap between Qwen3 8B and 4B (including the official versions) is only about 3 to 4 percentage points on all benchmarks except GPQA-Diamond/Main. In Figure 5, although SuperGPQA achieves an improvement of 4 percentage points, the visual distinction appears less pronounced due to the extensive range of the y-axis, spanning from 45% to 90%.
> - For MMLU, our current results are already near saturation. Prior work has shown that MMLU contains labeling errors that lead to underestimation of a model's true capability [1]. Consequently, most leading models achieve around 90% on MMLU rather than 100%. For example, GPT-4.1 reports 90.2% [2]. Our Qwen3-8B model, trained on only 0.3 million of our synthesized samples, already reaches 86.43%, leaving limited room for further improvement.
> - Therefore, more difficult datasets (GPQA-Diamond and SuperGPQA) are not yet saturated and are thus more sensitive to increases in data scale, whereas relatively simpler or near-ceiling benchmarks (MMLU and MMLU-Pro) exhibit insensitivity to incremental data.
>
> ---
>
> **References**
>
> [1] Vendrow, J., Vendrow, E., Beery, S., & Madry, A. (2025). Do Large Language Model Benchmarks Test Reliability? arXiv preprint arXiv:2502.03461.
>
> [2] OpenAI. (2025). GPT-4.1 Model Card.

---

> ### Author Response · Authors · 2025-11-24
> **Response to Reviewer VTR2 (3/4)**
>
> **For Weakness 3.2**
>
> >Weakness 3.2: In Table 5, the ablation study shows that removing Design Logic (w/o Design Logic) results in only minor performance drops. Other ablation variants also show limited improvements?
>
> ### Regarding "removing Design Logic (w/o Design Logic) results in only minor performance drops"
> 1. Even the ablation variants such as "w/o Design Logic" benefit from the upstream components of our pipeline, including data filtering and other preprocessing steps. The book/web corpora and the question exemplars used are high quality. As a result, the data produced by these naive variants remain of inherently strong quality.
>
> 2. For ease of verification, the current ablations in Table 5 were conducted on a smaller book subset containing 304,181 samples, which naturally reduces the achievable upper bound compared with experiments that use the full dataset. We therefore further evaluated the "w/o Design Logic" variant on the full book corpus (3.04 million samples). The results are shown in the table below. The performance degradation of "w/o Design Logic (Full)" is notably significant. Specifically, on the GPQA-Diamond benchmark, the performance gap between "w/o Design Logic (Full)" and the model employing "Design Logic" reaches **5** percentage points, providing stronger evidence for the advantage of employing design logic.
>
> | Method                         | MMLU   | MMLU-Pro | GPQA-Diamond      | GPQA-Diamond (CoT-SC) | SuperGPQA |
> |-------------------------------|--------|----------|--------------------|-------------------------|-----------|
> | DESIGNER (Full)               | 87.53  | 76.69    | 69.39±1.87         | 73.74                  | 50.57     |
> | w/o Design Logic (Full)       | 86.51  | 75.54    | 64.10±2.81         | 67.17                  | 48.46     |
>
> 3. The value of design logic lies not only in its effect on final performance but also in its controllable, abstractable, and generalizable synthesis mechanism. We verify that the synthesized questions adhere to the corresponding design logic. We used GPT-5.1 and a specifically designed prompt (Figure 15) to check the logical consistency of 10,000 randomly sampled data points from our synthesized dataset. The proportion of synthesized questions conforming to the designed logic (where GPT-5.1 outputted "Yes") is **84.69%**. The guidance of the design logic is crucial for ensuring the difficulty and diversity of the synthesized questions. Section 4.1 (Difficulty Analysis) shows that our design logics allow precise control over question difficulty, enabling the generation of highly challenging questions far beyond baseline datasets and commonly used benchmarks. Section 4.2 (Diversity Analysis) shows that design-logic-guided synthesis achieves higher scores on all five diversity metrics compared with baseline datasets.
>
> 4. Although we cannot release our full question bank due to copyright constraints, we can release the design logic library to facilitate community research on synthetic data generation. The design logic is a form of reusable meta-knowledge that abstracts the underlying reasoning structure, enabling LLMs to generate new questions with the same complex reasoning patterns from entirely different source texts. Due to the file size limitation of the supplementary material, we have uploaded our extracted complete design logic library and a randomly sampled set of 3,000 instances from our synthesized complete dataset as an example of the dataset that we will release. These samples contain fields such as questions, long CoT responses, and labels for discipline, difficulty, and question type.
>
>
> ### Regarding "Other ablation variants also show limited improvements"
> The other ablation variants (w/o Coarse Ranking and w/o Fine Ranking) show only limited gains because they already benefits from our method (design logic) and data filtering, which yield high-quality data. Removing either the coarse or fine ranking stage only disables part of the mechanism for selecting the most suitable design logic, resulting in the use of a suboptimal design logic. Consequently, the expected impact is naturally small. We are constrained by the cost and time required for data synthesis, which prevents us from scaling these two ablations further because each run requires generating a new set of 3.04 million questions and the corresponding long CoT responses. However, the current results on approximately 300k samples already demonstrate consistent gains from using the full DESIGNER method.

---

> ### Author Response · Authors · 2025-11-24
> **Response to Reviewer VTR2 (4/4)**
>
> **For Weakness 4**
>
> >Weakness 4: The data efficiency of the synthesis process is unclear. For example, during data filtering, what proportion of data was discarded? Understanding this would help assess the trade-off between data quality and synthesis cost.
>
> Thank you for raising this important question. We clarify the data efficiency and filtering ratios for each stage of our Data Curation pipeline (Question Bank, Book Corpus, and Web Corpus) below. In summary, our filtering is primarily driven by compute budget constraints rather than data quality issues, indicating that a vast amount of high-quality data remains available for scaling up.
>
> (1) Question Bank Processing
>
> The raw pool of 150 million questions aligns with the subject distribution in Table 6, with a difficulty ratio (Very Hard:Hard:Medium:Easy) of approximately 1:3:20:76. Labeling for subject and difficulty involves no filtering; theoretically, any question can yield design logic. We selected a subset of 132,409 questions for logic extraction purely based on a compute budget. To optimize the difficulty distribution within this subset, we clustered questions by subject and applied a stratified sampling ratio of 3:2:1 (Very Hard:Hard:Medium). Note that the unselected questions in these clusters remain valid for future extraction; they were not discarded due to quality concerns.
>
> (2) Book Corpus Processing
>
> Starting with ~40 million book segments (5k words each), Readability Filtering removed less than 10% of the data, primarily targeting OCR errors and garbled text. We then used the FineWeb-Edu classifier to assess educational value: approximately 60% of the data scored >= 2. Given the abundance of high-quality segments, we prioritized the top-tier data. Due to compute limits, we used only ~3 million segments to synthesize 3 million questions. The vast majority of the high-quality book corpus remains available for further synthesis.
>
> (3) Web Corpus Processing
>
> We found the FineWeb-Edu classifier too strict for certain subjects (low recall). Instead, we utilized LLMs and prompt to identify texts containing disciplinary reasoning. Only about 2.6% of raw web segments are judged to contain reasoning content (score ≥3). While this proportion reflects the naturally low density of reasoning content on the open web, the massive scale of the source corpus meant that even this small fraction yielded a massive volume of qualified texts. Similar to the book corpus, we only used a subset of these filtered texts for synthesis due to budget constraints. we synthesize 1.66M questions (DLR‑Web) from 1.66M web segments, leaving many additional reasoning‑qualified texts unused.
>
> (4) Design-Logic-Guided Data Synthesis
>
> Logic Extraction: This is a 1-to-1 process with no data filtering.
>
> Logic Deduplication: We remove design logics that are semantically redundant within a discipline via graph‑based clustering. We retained approximately 95% of the extracted design logic, attributing this high retention rate to our effective question selection strategy.
>
> Question Deduplication & Decontamination: For each selected book or web segment, we generate exactly one reasoning question. The subsequent Question Deduplication and Decontamination stage removes fewer than 1,000 questions out of ~4.7M synthesized questions (DLR‑Book + DLR‑Web), i.e., <0.03%. This demonstrates that our material-driven synthesis process rarely generates highly similar questions. Moreover, because our method does not expand from any seed question pool, it almost never produces text that is close to the questions contained in existing benchmarks.
>
> ---
>
> Finally, we would like to express our gratitude once again for the comprehensive suggestions you have provided to help improve our paper. We hope we have addressed your concerns.

---

### Official Review · Reviewer_mg22 · 2025-10-31

**Soundness:** 3
**Presentation:** 3
**Contribution:** 2
**Rating:** 2
**Confidence:** 4

**Summary:**

This paper introduces DESIGNER, a novel pipeline for synthesizing large-scale, multidisciplinary reasoning data by leveraging "design logic"—a structured abstraction of human educators' question-creation process. The method reverse-engineers design logics from existing questions. It matches them with raw text corpora (books and web) to generate challenging questions across 75 disciplines, resulting in two datasets: DLR-Book (3.04 million questions) and DLR-Web (1.66 million questions). Experiments demonstrate that models fine-tuned on these datasets achieve significant improvements in multidisciplinary reasoning, even surpassing official instruction-tuned counterparts. The work addresses data scarcity in complex reasoning domains but raises questions about scalability and quality control.

**Strengths:**

The step-by-step pipeline is highly systematic, enabling large-scale query data generation with strong operability and reproducibility. The clear phases—from data curation and design logic extraction to question synthesis—provide a structured framework that can be easily adapted or extended by other researchers. This practicality enhances the method's value for real-world applications.

**Weaknesses:**

1) The work resembles a complex engineering effort with multiple sub-tasks (e.g., discipline labeling, difficulty classification) pieced together, which may lack the novelty.

2) The pipeline relies heavily on prompt engineering (PE) at multiple stages (e.g., discipline classification using Figure 7, difficulty classification using Figure 8), but lacks rigorous quality assessments for each step. For instance, the discipline labels and difficulty scores are derived from LLM judgments without validation against ground-truth metrics or inter-annotator agreement. The difficulty classification, in particular, depends on only three examples in Figure 8, which may not capture nuanced criteria for "Very Hard" questions. Consequently, the claim in Section 4.1—that the synthesized data is more difficult based on LLM-labeled distributions—is less convincing without evidence of labeling reliability.

3) The focus on difficulty and diversity (Sections 4.1–4.2) overlooks critical aspects of data usability and accuracy, such as whether questions have unique answers or sufficient conditions to derive solutions. No analysis is provided on logical consistency or answer correctness, which are fundamental for reasoning data quality. This gap undermines the practical utility of the synthesized datasets.

Overall, the paper presents a scalable data synthesis pipeline with practical benefits, but the engineering-heavy approach, unvalidated quality controls, and overlooked usability issues limit its conceptual impact. Addressing these concerns could elevate the contribution.

**Questions:**

1) Will the synthesized datasets (DLR-Book and DLR-Web) be openly released to facilitate reproducibility and broader research?
2) In Line 400 (Section 5.5), the claim that "book corpora are of higher quality than web corpora" is presented as a widely acknowledged consensus but lacks references.
3) There appears to be a discrepancy in MMLU-Pro scores: Table 4 reports 68.4 for DLR-Book, while Table 5 reports 68.1 for the full DESIGNER method. Is this a typo? Additionally, Table 3 shows that DLR-Web+Book yields the best performance, yet Tables 4 and 5 use DLR-Book or ablated versions for comparisons. Justifying why the optimal dataset wasn't used in later analyses would improve consistency.

---

> ### Author Response · Authors · 2025-11-24
> **Response to Reviewer mg22 (1/4)**
>
> We appreciate that you find our systematic pipeline enables large-scale generation with strong reproducibility, our structured framework adaptable for other researchers, and our method practical for real-world applications. We have carefully studied your suggestions and further refined our paper based on your advice. We hope our following response will alleviate your concerns.
>
> ---
>
> **For Weakness 1**
>
> >Weakness 1: The work resembles a complex engineering effort with multiple sub-tasks (e.g., discipline labeling, difficulty classification) pieced together, which may lack the novelty.
>
> 1. Existing methods typically fall into two categories: query-centric (constrained by the coverage of the seed pool and the inherent model biases) or doc-centric (lacking control over difficulty). By reverse-engineering the meta-knowledge of human educators, we propose a fundamentally new question synthesis method guided by "design logic''. This approach enables the generation of truly complex, multi-step reasoning questions from raw text by providing the structured, reusable, and abstract control over difficulty and diversity that prior doc-centric methods lacked, which often restricted them to simple factual recall. We are the first to introduce reusable "design logic" as structured meta-knowledge to guide large-scale cross-domain synthesis. The guidance of the design logic is crucial for ensuring the difficulty and diversity of the synthesized questions.
> 2. Other reviewers do not consider our method to lack novelty. For example, Reviewer VsnW noted "The proposed DESIGNER data synthesis pipeline is novel and well-motivated. The structured process human educators use to construct complex and insightful questions." and Reviewer TpGs stated "Design logic direction seems like a novel and scalable method for condensing multi-domain learnings in a reusable manner."
> 3. To obtain high-quality data, it is necessary to apply data filtering, discipline labeling, difficulty classification, and related processing to the raw materials. These steps are essential for ensuring the quality of the synthetic data; they are **not related** to the **novelty** of the proposed method.
> 4. Moreover, our paper provides a detailed illustration of a real data-engineering pipeline that can synthesize data capable of substantially improving LLMs' reasoning ability. This detailed pipeline is precisely what is often absent from the academic research community. Even open-source models like Qwen generally do not disclose the precise steps for their training data preparation, and such information is certainly not available for closed-source models such as GPT and Gemini.

---

> ### Author Response · Authors · 2025-11-24
> **Response to Reviewer mg22 (2/4)**
>
> **For Weakness 2**
> >Weakness 2: The pipeline relies heavily on prompt engineering (PE) at multiple stages (e.g., discipline classification using Figure 7, difficulty classification using Figure 8), but lacks rigorous quality assessments for each step. For instance, the discipline labels and difficulty scores are derived from LLM judgments without validation against ground-truth metrics or inter-annotator agreement. The difficulty classification, in particular, depends on only three examples in Figure 8, which may not capture nuanced criteria for "Very Hard" questions. Consequently, the claim in Section 4.1—that the synthesized data is more difficult based on LLM-labeled distributions—is less convincing without evidence of labeling reliability.
>
> 1. We would like to clarify that Figures 7, 8, and 9 all utilize few-shot prompts. Their primary function is to standardize the model's output format to facilitate label extraction, and they do not affect the model's accuracy in determining the labels. Cheng et al. [1] pointed out that the main function of few-shot exemplars is to "align the output format," ensuring it meets human expectations, and this function operates independently of the model's inherent reasoning capability. Furthermore, Min et al. [2] demonstrated that even when the labels within the few-shot examples are randomly substituted, the model's final performance shows almost no degradation. These indicate that the quality of the exemplars we used does not influence the conclusions we draw. For a labeling task of this nature, the performance of our selected model, Qwen3-30B-A3B-Instruct-2507, is sufficiently strong. LLMs are intrinsically capable of determining labels such as discipline and difficulty for a given question; thus, the concern that the classification "depends on only three examples in Figure 8, which may not capture nuanced criteria for 'Very Hard' questions" is mitigated.
>
> 2. More importantly, the higher difficulty of our synthesized data is established through comparative analysis. As stated in Section 4.1 of the paper, "To facilitate an intuitive comparison of difficulty levels, we further applied the same labeling procedure to several commonly used benchmarks." We applied the same model and prompt to benchmark datasets to establish comparable difficulty standards. Moreover, the observed differences in difficulty between benchmarks like GSM8K and GPQA-Diamond align with the generally accepted understanding. For instance, GSM8K primarily consists of simple questions, with almost no "Hard" or "Very Hard" questions, whereas GPQA-Diamond shows a significantly higher proportion of "Hard" questions. Under the same scoring criteria, comparing the difficulty distribution of our dataset with commonly used benchmarks and baselines shows that our dataset contains a substantially larger proportion of high-difficulty questions.
>
> 3. We utilized GPT-5.1 (the prompts are provided in Figure 17 - 19) to check the correctness of every question's discipline, difficulty, and question type labels for a random sample of 10,000 data points drawn from our synthesized dataset. The proportions of questions possessing appropriate discipline, difficulty, and question type labels were found to be 90.14%, 96.53%, and 91.22%, respectively. (See Section 4.4 of the paper).
>
> 4. The ultimate validation of data difficulty is the **performance of the model trained on our data**. Models trained on our synthesized data achieve significantly larger gains on highly complex, human-expert-level benchmarks such as GPQA-Diamond, even surpassing their official final models that have undergone the full post-training process. This substantiates that our synthesized questions are effective in improving the model's reasoning capabilities on very difficult questions.
>
> ---
>
> **References**
>
> [1] Cheng, X., Pan, C., Zhao, M., Li, D., Liu, F., Zhang, X., ... & Liu, Y. (2025). Revisiting Chain-of-Thought Prompting: Zero-shot Can Be Stronger than Few-shot. EMNLP 2025 Findings.
>
> [2] Min, S., Lyu, X., Holtzman, A., Artetxe, M., Lewis, M., Hajishirzi, H., & Zettlemoyer, L. (2022, December). Rethinking the Role of Demonstrations: What Makes In-Context Learning Work?. EMNLP 2022.

---

> ### Author Response · Authors · 2025-11-24
> **Response to Reviewer mg22 (3/4)**
>
> **For Weakness 3**
>
> >Weakness 3: The focus on difficulty and diversity (Sections 4.1–4.2) overlooks critical aspects of data usability and accuracy, such as whether questions have unique answers or sufficient conditions to derive solutions. No analysis is provided on logical consistency or answer correctness, which are fundamental for reasoning data quality. This gap undermines the practical utility of the synthesized datasets.
>
> ### 1. Regarding "whether questions have unique answers"
>
> We would like to clarify that our framework does not mandate that questions possess a unique answer. The majority of the questions we synthesize are complex problem-solving tasks and open-ended questions. Furthermore, our objective is to create a multi-disciplinary synthesis dataset. Consequently, many fields such as Humanities and Social Sciences, Applied and Professional Fields, and Arts contain open-ended questions that do not have a single, fixed solution.
>
> ### 2. Regarding "whether questions have sufficient conditions to derive solutions"
>
> We designed a prompt (presented in the newly added Figure 14 in the paper) and utilized GPT-5.1 to evaluate this specific condition on 10,000 randomly sampled data points from our synthesized dataset. The proportion of questions that are complete and answerable (where GPT-5.1 outputted "Yes") is **96.70%**. This result demonstrates that almost all questions are solvable, with only a very small minority being unanswerable.
>
> ### 3. Regarding "logical consistency"
>
> We used GPT-5.1 and a specifically designed prompt (presented in the newly added Figure 15 in the paper) to check the logical consistency of 10,000 randomly sampled data points from our synthesized dataset. The proportion of synthesized questions conforming to the designed logic (where GPT-5.1 outputted "Yes") is **84.69%**.
>
> We have incorporated the above results into Section 4.4.
>
> ### 4. Regarding "answer correctness"
>
> We wish to clarify that the **response generation is not the primary focus of our paper**. Our main objective and proposed method concentrate on synthesizing **challenging and diverse multidisciplinary questions**. We believe that generating high-quality questions is often more critical than generating answers. Provided a high-quality question, the response can be generated using any model, including current state-of-the-art models such as GPT-5.1 and Gemini 3, or even more powerful future models, to attain higher response accuracy.
>
> The ultimate criterion for data quality is model performance. If the usability and accuracy of the data were insufficient, the model would typically exhibit increased hallucination or degraded performance after SFT. However, models trained on our data achieve significantly larger gains on highly complex, human-expert-level benchmarks like GPQA-Diamond, even surpassing their official final models that have undergone the full post-training process. The data scaling curve in Figure 5 indicates a continuous and stable improvement in model performance as the data volume increases. These provide direct evidence that our data quality is sufficiently high.
>
> Finally, we would like to underscore that our proposed methodology does not conflict with any quality control approach; subsequent strategies for quality control and data filtering can be applied to the existing synthesized data we provide. However, this is not the focus of our current work but is rather intended for future research.
>
> ---
>
> **For Question 1**
>
> >Question 1: Will the synthesized datasets (DLR-Book and DLR-Web) be openly released to facilitate reproducibility and broader research?
>
> 1. We have explicitly stated in the Reproducibility Statement section of the paper that: "To facilitate reproducibility and independent verification, we will release a subset of our synthesized data that does not involve legal compliance concerns or potential conflicts of interest. The released data will include our synthesized questions, reference answers, long CoT responses, the corresponding source corpora and design logic, as well as associated labels (e.g., discipline, difficulty, question type), enabling reproduction and validation of our method's effectiveness and the quality of the synthesized data. Furthermore, we will release a design logic library that contains all design logics extracted in this study, together with discipline, difficulty, and question type labels, enabling the research community to directly match these design logics to their own raw documents and synthesize new questions that exhibit the same complex reasoning."
> 2. Due to the file size limitation of the supplementary material, we have uploaded our extracted complete design logic library and a randomly sampled set of 3,000 instances from our synthesized complete dataset as an example of the dataset that we will release. These samples contain fields such as questions, long CoT responses, and labels for discipline, difficulty, and question type.

---

> ### Author Response · Authors · 2025-11-24
> **Response to Reviewer mg22 (4/4)**
>
> **For Question 2**
>
> >Question 2: In Line 400 (Section 5.5), the claim that "book corpora are of higher quality than web corpora" is presented as a widely acknowledged consensus but lacks references.
>
> Thank you for pointing out the need for references. The consensus that book corpora are of higher quality than web corpora is supported by foundational studies. We haved cited the following papers in the revised version:
>
> - GPT-3 [3]: In Section 2.2 (arXiv version), the authors explicitly classify their book corpora as "known high-quality reference corpora" added to augment Common Crawl. The authors explicitly up-sampled their book corpora during training, stating that "datasets we view as higher-quality are sampled more frequently" than Common Crawl.
>
> - The Pile [4]: In Section 2.3, the authors justify the inclusion of the Books3 dataset by noting that "books are invaluable for long-range context modeling research and coherent storytelling", qualities often lacking in fragmented web scraps.
>
> - Gopher [5]: In Appendix A.3.1, DeepMind's ablation study on dataset weights explicitly reports: "We find that using a high proportion of Books reduces the loss on LAMBADA". Furthermore, in Section 4.2, they attribute their model's strong performance on knowledge-intensive tasks to "the heavy use of book data in MassiveText".
>
> However, our experiments in Section 5.5 show that, for our method, the performance gap induced by the two source corpora of differing quality is negligible. This result indicates that our method is robust to variations in source quality and can reliably generate high-quality questions even when applied to lower-quality corpora.
>
> ---
>
> **References**
>
> [3] Brown, T. B., Mann, B., Ryder, N., Subbiah, M., Kaplan, J., Dhariwal, P., ... & Amodei, D. (2020). Language Models are Few-Shot Learners. arXiv preprint arXiv:2005.14165.
>
> [4] Gao, L., Biderman, S., Black, S., Golding, L., Hoppe, T., Foster, C., ... & Leahy, C. (2020). The pile: An 800gb dataset of diverse text for language modeling. arXiv preprint arXiv:2101.00027.
>
> [5] Rae, J. W., Borgeaud, S., Cai, T., Millican, K., Hoffmann, J., Song, F., ... & Irving, G. (2021). Scaling language models: Methods, analysis & insights from training gopher. arXiv preprint arXiv:2112.11446.
>
> ---
>
> **For Question 3**
>
> >Question 3: There appears to be a discrepancy in MMLU-Pro scores: Table 4 reports 68.4 for DLR-Book, while Table 5 reports 68.1 for the full DESIGNER method. Is this a typo? Additionally, Table 3 shows that DLR-Web+Book yields the best performance, yet Tables 4 and 5 use DLR-Book or ablated versions for comparisons. Justifying why the optimal dataset wasn't used in later analyses would improve consistency.
>
> 1. The values 68.4 and 68.1 that you mentioned never appear in our paper (including Table 4 and Table 5), so this is likely a misreading.
>
> 2. Because the sizes of the compared datasets differ substantially and most baseline datasets are smaller than ours, we must instead balance the smallest dataset size and the available resources. As stated in lines 391–393, "To ensure a fair comparison, given the varying sizes of the original datasets, we randomly sampled an equal number of 304,181 instances from each dataset." By restricting all methods to a single high-quality domain (the book corpus) and controlling the data scale (sampling 304,181 instances), we eliminate confounding variables introduced by mixing multiple data sources. This design allows us to attribute the observed performance gains strictly to the proposed synthesis method rather than to differences in data source diversity.
>
> ---
>
> Finally, we would like to extend our gratitude for all the suggestions you have provided to help enhance our paper. We have incorporated these details into our paper to further strengthen our work. We hope our response alleviates your concerns.

---

### Official Review · Reviewer_VsnW · 2025-11-01

**Soundness:** 4
**Presentation:** 4
**Contribution:** 3
**Rating:** 8
**Confidence:** 2

**Summary:**

This paper identifies the primary bottleneck of improving the complex, multi-step reasoning capabilities of Large Language Models (LLMs) as a scarcity of large-scale, high-quality, and diverse training data, especially for specialized, university-level subjects. To solve this, the paper introduces **DESIGNER**, a novel data synthesis pipeline. The core innovation is the concept of "Design Logic", which the authors define as a form of "meta-knowledge" that abstracts the structured process human educators use to construct complex and insightful questions.

The paper's primary contributions are:
- The **DESIGNER** pipeline, centered on the novel "Design Logic" concept.
- The two resulting multi-million-question datasets, DLR-Book and DLR-Web, whose analysis shows significantly more difficult and semantically diverse than existing baseline datasets.
- Extensive validation through Supervised Fine-Tuning (SFT) on Llama3 and Qwen3 model families. The key result is that base models trained on the proposed datasets substantially outperform existing datasets.

**Strengths:**

1. The proposed **DESIGNER** data synthesis pipeline is novel and well-motivated. The structured process human educators use to construct complex and insightful questions.
1. Using this pipeline, the authors created two new, large-scale reasoning datasets: **DLR-Book** (3.04 million questions) and **DLR-Web** (1.66 million questions), which benefit the community to improve existing models.
1. Detailed analyses prove with quantitative metrics that the DLR datasets are more difficult and more semantically diverse, validating the effectiveness of the proposed **DESIGNER** pipeline.
1. Applying SFT on the DLR datasets, existing models outperform their official versions and gain performance improvement across all reasoning benchmarks.
1. Throughout ablation studies justify the effectiveness of the **DESIGNER** pipeline

**Weaknesses:**

1. The pipeline's success relies on massive proprietary assets. Though the authors state they will release a subset of the final synthesized data, this still limits the community's ability to fully reproduce the pipeline or build upon the design logic library.
1. The pipeline itself depends on existing very large, capable models (e.g., Qwen3-30B, DeepSeek-R1-0528), and this means massive computational resources are required to apply this pipeline.
1. The pipeline only creates large-scale datasets, and these datasets are only validated on very large models, missing validation on small corpora and small models.

**Questions:**

This paper is quite solid, detailed, and well-written, and I only have a few questions:
1. In Figure 4, we can see that DLR-Book contains more than half of the data categorized as "Other" (and more than 1/3 on DLR-Web). Is this beneficial? If so, how do these benefit the model? If not, can we drop them?
1. Can the **DESIGNER** pipeline apply to open-source raw data or existing datasets?
1. How many GPU hours have been spent on creating the DLR datasets?

---

> ### Author Response · Authors · 2025-11-24
> **Response to Reviewer VsnW (1/2)**
>
> We sincerely appreciate your insightful comments and high evaluation. We are encouraged that you find our DESIGNER pipeline novel and well-motivated, our resulting datasets valuable and empirically validated, and our detailed analyses effective.
>
> ---
>
> **For Weakness 1 & Question 2**
>
> > Weakness 1: The pipeline's success relies on massive proprietary assets. Though the authors state they will release a subset of the final synthesized data, this still limits the community's ability to fully reproduce the pipeline or build upon the design logic library.
> >
> > Question 2: Can the DESIGNER pipeline apply to open-source raw data or existing datasets?
>
> Proprietary assets originate from two sources: the question bank used to extract design logics and the raw documents used to synthesize questions:
> 1. Regarding the question bank, most questions in our question bank were cleaned from publicly available web data, but due to copyright constraints, we cannot release the questions themselves. In fact, our raw question bank is not of high quality, and most questions are not challenging. As described in Section 2.1, "We annotate over 150 million questions from the proprietary question bank," and after multi-stage filtering, we retain "a curated set of 132,409 questions for design logic extraction." Any question bank, regardless of its initial quality, can be processed using the filtering pipeline in Section 2.1 to obtain a high-quality and diverse subset for extracting design logics. Our method is therefore applicable to arbitrary question banks, and it can be directly applied to publicly available datasets (e.g., `allenai/SciQ` and `MegaScience/MegaScience` on Hugging Face).
> 2. We will release a design logic library that contains all design logics extracted in this study, together with difficulty, question type, and discipline labels, enabling the research community to directly match these design logics to their own raw documents and synthesize new questions that exhibit the same complex reasoning.
> 3. For the original documents, our DLR-web data are drawn from the FineFineWeb dataset, which is open-source raw data.
> 4. We provide all complete and detailed data processing steps used for data synthesis in the main paper and the appendix. For the final synthesized data, the released data will include our synthesized questions, reference answers, long CoT responses, the corresponding source corpora and design logic, as well as associated labels (e.g., discipline, difficulty, question type), enabling reproduction and validation of our method's effectiveness and the quality of the synthesized data.
> 5. Due to the file size limitation of the supplementary material, we have uploaded our extracted complete design logic library and a randomly sampled set of 3,000 instances from our synthesized complete dataset as an example of the dataset that we will release. These samples contain fields such as questions, long CoT responses, and labels for discipline, difficulty, and question type.
>
> ---
>
> **For Weakness 2**
>
> > Weakness 2: The pipeline itself depends on existing very large, capable models (e.g., Qwen3-30B, DeepSeek-R1-0528), and this means massive computational resources are required to apply this pipeline.
>
> 1. Our proposed methodology is compatible with LLMs of any scale. However, to ensure the synthesis of ultra-challenging questions and to maximize data quality, we utilized the most powerful available LLMs, as objective is to contribute a dataset of the highest quality to the community. As stated in the response to the previous question, we will open-source the synthesized data. By incurring the high cost of a single synthesis process, we aim to provide the community with a reusable library of design logics and a high-quality training dataset.
> 2. Moreover, we anticipate that the size and capability of LLMs, as well as the corresponding hardware capacity (GPU), will continue to expand. Models that appear large today will likely no longer be considered large in the near future. If synthetic data were generated using very small models at present, the long-term utility of such data would decrease rapidly as model scales continue to grow.

---

> ### Author Response · Authors · 2025-11-24
> **Response to Reviewer VsnW (2/2)**
>
> **For Weakness 3**
>
> > Weakness 3: The pipeline only creates large-scale datasets, and these datasets are only validated on very large models, missing validation on small corpora and small models.
>
> 1. Our dataset has been validated on open-source models of multiple scales, including Llama-3.2-3B, Llama-3.1-8B, Qwen3-4B, and Qwen3-8B. The 3B and 4B models are generally considered relatively small, yet our data yields substantial performance gains even on these models and surpasses their official final models that have undergone the full post-training process. This confirms the generalization capability of our dataset.
> 2. Our synthetic dataset is designed to address multi-disciplinary reasoning challenges that inherently require models of sufficient scale to acquire cross-domain reasoning competence and to execute long CoT reasoning.
>
> ---
> **For Question 1**
>
> > Question 1: In Figure 4, we can see that DLR-Book contains more than half of the data categorized as "Other" (and more than 1/3 on DLR-Web). Is this beneficial? If so, how do these benefit the model? If not, can we drop them?
>
> Because visualizing all 75 disciplines in a pie chart would require 75 distinct colors and legends, which is impractical, we instead show the 10 most representative disciplines and aggregate the remaining ones into the "Other" category. We have further refined the description in Section 4.3 as follows: "For visualization, we highlight the ten most dominant disciplines within each dataset and those most representative of broader academic categories, while aggregating the remaining disciplines into the gray category 'Other'." The detailed number of questions per discipline in our dataset is provided in Appendix C.
> The inclusion of this large "Other" category is highly beneficial, serving two key purposes:
> 1. Multidisciplinary Robustness: These 65 disciplines encompass a wide range of fields, including Applied and Professional Fields (e.g., Clinical Medicine, Law), Humanities, and Arts. By including them, we ensure the model learns to reason across a truly broad disciplinary spectrum, preventing overfitting to a few dominant STEM subjects, thereby enhancing its general and robust reasoning capability.
> 2. Distribution Balance: As shown in Figure 4, many existing multidisciplinary datasets are heavily skewed toward a few disciplines, such as mathematics, leading to a highly imbalanced distribution across disciplines. The broad data distribution in our dataset can provide richer multi-disciplinary knowledge and, in particular, improve the model's performance on underrepresented disciplines.
>
> ---
>
> **For Question 3**
>
> > Question 3: How many GPU hours have been spent on creating the DLR datasets?
>
> The synthesis process for the entire DLR-Book and DLR-Web datasets (excluding model training experiments) utilized 128 H20 GPUs for approximately three weeks, corresponding to about 128 × 24 × 7 × 3 = 64,512 H20 GPU hours.
>
> ---
>
> Finally, we would like to extend our gratitude once again for all the suggestions you have provided and for the time you have devoted to our work.

---

> > ### Comment · Reviewer_VsnW · 2025-11-25
> >
> > Thank you. All my concerns are addressed. I will keep my score. Good luck :)

---

> ### Author Response · Authors · 2025-11-26
> **Thank you, Reviewer VsnW**
>
> Dear Reviewer VsnW,
>
> We sincerely thank you for your positive feedback and your confirmation that our responses have fully addressed your concerns. We are very grateful for your time and constructive comments throughout the review process. Your support is of great importance to us.
>
> Best regards,
>
> The Authors

---

### Official Review · Reviewer_TpGs · 2025-11-02

**Soundness:** 2
**Presentation:** 3
**Contribution:** 2
**Rating:** 4
**Confidence:** 4

**Summary:**

The paper proposes Designer, a data-synthesis pipeline that extracts reusable design logics (structured problem-solving flow charts) from a large question bank. Using filtered snippets form existing text corpora, the system coarse-retrieves several candidate logics, then has an LLM select one to use, and synthesizing snippet and logic to form a new question plus a short reference answer. A separate thinking model produces long chain-of-thought (CoT) responses. To test this pipeline, popular base models are SFT'd on these synthetic examples. They find that Designer SFT outperforms the instruciton-tuned variants of the base models. Several ablations are performed, including analysis of the corpora quality and the key components of the design logic process. The produced training data is also compared with identically-sized subsets of other synthetic datasets, finding the DLR-trained models consistently outperform the other 4 baselines.

**Strengths:**

- Design logic direction seems like a novel and scalable method for condensing multi-domain learnings in a reusable manner.
- Resultant datasets yields solid improvements over other synthetic pre-training data methods.
- Paper is well written and easy to follow.

**Weaknesses:**

My biggest concerns are with the use of a proprietary question bank, and also general concerns with the impact of the design logics, versus just having relevant and high-quality data:
- For the design logic process, I like the idea of having a static design logics bank, however, it seems the design logics themselves don't have a massive impact on performance; the 'w/o Design Logic' ablations note fairly small gains from using design logics (as opposed to just providing examples from the question bank). I'd love to see more extensive analysis into the design logic creation as this is the most novel component of this work.
- Given the strong dependency on the question bank, I am concerned about its proprietary nature. What happens if we don't have a carefully-crafted question bank? Is the Designer process is only pertinent if we have a challenging question bank available? If this process is largely reproducible with public data, then it would be ideal to publish these results.
- No comparison with other simple SFT baselines. For example, what are the base model results (no SFT), and what if you perform SFT on the instruction-tuned variants of the baseline models? It seems these results would paint a clearer picture.

**Questions:**

- **Significance of question bank**: Can you provide any results using only public / reproducible data? A key distinction between your work and others is the difficulty of the generated questions (Figure 3). Is it possible your data is simply more challenging due to this question bank? Also, if difficulty is a large factor, could one simply just upsample hard questions in other dataset to match the distribution of yours?
- **Additional SFT baselines**: Can you provide the raw base model results, and also Instruction-tuned base model + DLR SFT?
- **Analysis of synthetic generated responses**: Does the long-CoT Qwen3-235B responses ever disagree with the DeepSeek-R1 answer? If so, do you filter these examples out? Also, have you explored what happens if you include the corresponding design logics as problem-solving guidance (e.g. "for these types of problems some things to think about are A -> B -> C")
- **Contribution of design logics for question generation** -- again, the ablation in Table 5 notes that the gains from design logics are fairly minimal; it appears that the retrieved examples alone (w/o Design Logic) perform well, so could this mean that design logics are perhaps just useful as a structured and abstracted method for indexing a large question bank?

Small writing suggestion: The caption in Figure 6 ('source corpus quality') is ambiguous until you look at Section 5.5 (which discusses the acknowledged differences).

---

> ### Author Response · Authors · 2025-11-24
> **Response to Reviewer TpGs (1/5)**
>
> We appreciate that you like the idea of having a static design logics bank, find our design logic novel and scalable, consider our resultant datasets effective, and view our paper as well-written and easy to follow. We are also grateful for your detailed and helpful suggestions. We have carefully studied your suggestions and further refined our paper accordingly. We hope our following response will alleviate your concerns.
>
> ---
>
> **For Weakness 1 & Question 4**
>
> > Weakness 1: For the design logic process, I like the idea of having a static design logics bank, however, it seems the design logics themselves don't have a massive impact on performance; the 'w/o Design Logic' ablations note fairly small gains from using design logics (as opposed to just providing examples from the question bank). I'd love to see more extensive analysis into the design logic creation as this is the most novel component of this work.
> >
> > Question 4: Contribution of design logics for question generation -- again, the ablation in Table 5 notes that the gains from design logics are fairly minimal; it appears that the retrieved examples alone (w/o Design Logic) perform well, so could this mean that design logics are perhaps just useful as a structured and abstracted method for indexing a large question bank?
>
> 1. Even the ablation variants such as "w/o Design Logic" benefit from the upstream components of our pipeline, including data filtering and other preprocessing steps. The book/web corpora and the question exemplars used are high quality. As a result, the data produced by these naive variants remain of inherently strong quality.
>
> 2. For ease of verification, the current ablations in Table 5 were conducted on a smaller book subset containing 304,181 samples, which naturally reduces the achievable upper bound compared with experiments that use the full dataset. We therefore further evaluated the "w/o Design Logic" variant on the full book corpus (3.04 million samples). The results are shown in the table below. The performance degradation of "w/o Design Logic (Full)" is notably significant. Specifically, on the GPQA-Diamond benchmark, the performance gap between "w/o Design Logic (Full)" and the model employing "Design Logic" reaches **5** percentage points, providing stronger evidence for the advantage of employing design logic.
>
> | Method                         | MMLU   | MMLU-Pro | GPQA-Diamond      | GPQA-Diamond (CoT-SC) | SuperGPQA |
> |-------------------------------|--------|----------|--------------------|-------------------------|-----------|
> | DESIGNER (Full)               | 87.53  | 76.69    | 69.39±1.87         | 73.74                  | 50.57     |
> | w/o Design Logic (Full)       | 86.51  | 75.54    | 64.10±2.81         | 67.17                  | 48.46     |
>
> 3. The value of design logic lies not only in its effect on final performance but also in its controllable, abstractable, and generalizable synthesis mechanism. We verify that the synthesized questions adhere to the corresponding design logic. We used GPT-5.1 and a specifically designed prompt (Figure 15) to check the logical consistency of 10,000 randomly sampled data points from our synthesized dataset. The proportion of synthesized questions conforming to the designed logic (where GPT-5.1 outputted "Yes") is **84.69%**. The guidance of the design logic is crucial for ensuring the difficulty and diversity of the synthesized questions. Section 4.1 (Difficulty Analysis) shows that our design logics allow precise control over question difficulty, enabling the generation of highly challenging questions far beyond baseline datasets and commonly used benchmarks. Section 4.2 (Diversity Analysis) shows that design-logic-guided synthesis achieves higher scores on all five diversity metrics compared with baseline datasets.
>
> 4. Although we cannot release our full question bank due to copyright constraints, we can release the design logic library to facilitate community research on synthetic data generation. The design logic is a form of reusable meta-knowledge that abstracts the underlying reasoning structure, enabling LLMs to generate new questions with the same complex reasoning patterns from entirely different source texts.

---

> ### Author Response · Authors · 2025-11-24
> **Response to Reviewer TpGs (2/5)**
>
> **For Weakness 2 & Question 1**
> > Weakness 2: Given the strong dependency on the question bank, I am concerned about its proprietary nature. What happens if we don't have a carefully-crafted question bank? Is the Designer process is only pertinent if we have a challenging question bank available? If this process is largely reproducible with public data, then it would be ideal to publish these results.
> >
> > Question 1: Significance of question bank: Can you provide any results using only public / reproducible data? A key distinction between your work and others is the difficulty of the generated questions (Figure 3). Is it possible your data is simply more challenging due to this question bank?
>
> 1. We will release a design logic library that contains all design logics extracted in this study, together with difficulty, question type, and discipline labels, enabling the research community to directly match these design logics to their own raw documents and synthesize new questions that exhibit the same complex reasoning. Due to the file size limitation of the supplementary material, we have uploaded our extracted complete design logic library and a randomly sampled set of 3,000 instances from our synthesized complete dataset as an example of the dataset that we will release. These samples contain fields such as questions, long CoT responses, and labels for discipline, difficulty, and question type. As you noted in your strengths, "Design logic direction seems like a novel and scalable method for condensing multi-domain learnings in a reusable manner." With the released library, we believe this can be readily achieved in practice.
> 2. We would like to clarify that our approach does not rely on "the strong dependency on the proprietary / carefully-crafted / challenging question bank." Most questions in our question bank were cleaned from publicly available web data, but due to copyright constraints, we cannot release the questions themselves. In fact, our raw question bank is not of high quality, and most questions are not challenging. As described in Section 2.1, "We annotate over 150 million questions from the proprietary question bank," and after multi-stage filtering, we retain "a curated set of 132,409 questions for design logic extraction." Any question bank, regardless of its initial quality, can be processed using the filtering pipeline in Section 2.1 to obtain a high-quality and diverse subset for extracting design logics. Our method is therefore applicable to arbitrary question banks, and it can be directly applied to publicly available datasets (e.g., `allenai/SciQ` and `MegaScience/MegaScience` on Hugging Face).

---

> ### Author Response · Authors · 2025-11-24
> **Response to Reviewer TpGs (3/5)**
>
> **For Question 1**
>
> >Also, if difficulty is a large factor, could one simply just upsample hard questions in other dataset to match the distribution of yours?
>
> We are pleased to share our perspective on this point with you.
>
> 1. To eliminate the influence of disparate difficulty distributions, we mixed and upsampled data from four baseline datasets to construct a hybrid dataset whose difficulty ratio closely matches that of our DLR-Book dataset. This hybrid dataset contains 304,181 instances, consistent with the sample size used in the comparison in Section 5.3. The results are shown in the table below. The performance remains consistently lower than that obtained with our dataset across all benchmarks. This gap is likely due to limitations in the diversity and subject coverage of their datasets relative to ours.
>
> | Dataset                       | **MMLU** | **MMLU-Pro** | **GPQA-Diamond**      | **GPQA-Diamond (CoT-SC)** | **SuperGPQA** |
> |------------------------------|----------|--------------|-------------------------|-----------------------------|---------------|
> | OpenThoughts3                | 72.49    | 57.76        | 45.86±1.80             | 54.04                      | 39.70         |
> | Nemotron-Post-Training-v1    | 77.17    | 62.52        | 38.59±1.24             | 40.91                      | 42.03         |
> | WebInstruct (Full)           | 86.34    | 72.83        | 55.61±2.50             | 62.63                      | 45.37         |
> | NaturalReasoning             | 85.33    | 72.39        | 56.67±2.20             | 60.00                      | 43.38         |
> | Upsampled Baseline           | 86.11    | 73.02        | 56.97±2.35             | 60.10                      | 44.23         |
> | DLR-Web                      | 86.32    | 73.81        | 58.89±1.98             | 63.64                      | **47.23**     |
> | **DLR-Book**                 | **86.43**| **74.98**    | **60.35±1.93**         | **66.67**                  | 47.04         |
>
> 2. Our advantage does not stem solely from difficulty, but from broader diversity (Section 4.2) and a more uniform disciplinary distribution (Section 4.3). Most baseline datasets exhibit pronounced disciplinary imbalance (as shown in Figure 4), with heavy concentration in a few dominant areas such as math. Consequently, simply upsampling difficult questions cannot compensate for the lack of diversity and insufficient disciplinary coverage.
>
> 3.  Furthermore, our proposed methodology demonstrates superior efficiency in synthesizing challenging questions, with minimal generation of easy questions that offer limited utility for model training. As a direct result, models trained on our synthesized data achieve the most substantial performance gains on the most difficult benchmarks, such as GPQA-Diamond and SuperGPQA.

---

> ### Author Response · Authors · 2025-11-24
> **Response to Reviewer TpGs (4/5)**
>
> **For Weakness 3 & Question 2**
> >Weakness 3: No comparison with other simple SFT baselines. For example, what are the base model results (no SFT), and what if you perform SFT on the instruction-tuned variants of the baseline models? It seems these results would paint a clearer picture.
> >
> >Question 2: Additional SFT baselines: Can you provide the raw base model results, and also Instruction-tuned base model + DLR SFT?
>
> We appreciate the suggestion to include additional baselines to provide a more complete picture. Given the limited time and computational resources, and the fact that our dataset is particularly large and slow to train on, we opted to conduct this experiment on the most advanced and representative Qwen3 8B model. We have incorporated the following additional results in the revised version of the paper:
>
> | Model | MMLU | MMLU-Pro | GPQA-Diamond (Acc) | GPQA-Diamond (CoT-SC) | GPQA-Main (Acc) | GPQA-Main (CoT-SC) | SuperGPQA |
> | :--- | :---: | :---: | :---: | :---: | :---: | :---: | :---: |
> | Qwen3-8B-Base | 76.23 | 55.69 | 43.13±1.78 | 45.45 | 41.05±1.25 | 43.75 | 30.54 |
> | Qwen3-8B (Thinking Mode) | 85.85 | 73.62 | 59.44±2.53 | 60.61 | 57.95±1.47 | 59.38 | 47.52 |
> | Qwen3-8B-Base-SFT (DLR-Web) | 86.82 | 75.62 | 63.28±2.43 | 66.67 | 61.43±0.98 | 66.07 | 48.66 |
> | Qwen3-8B-Base-SFT (DLR-Book) | 87.53 | 76.69 | 69.39±1.87 | 73.74 | 65.07±0.98 | 68.30 | 50.57 |
> | Qwen3-8B-Base-SFT (DLR-Web+Book) | 87.60 | 76.72 | 71.01±2.33 | 75.76 | 65.40±1.05 | 69.20 | 50.90 |
> | Qwen3-8B-SFT-continued (DLR-Web) | 87.20 | 76.73 | 63.89±2.64 | 68.18 | 61.05±1.27 | 65.18 | 49.14 |
> | Qwen3-8B-SFT-continued (DLR-Book) | 88.07 | 77.94 | 70.35±2.05 | 74.24 | 66.83±1.14 | 69.64 | 51.27 |
> | Qwen3-8B-SFT-continued (DLR-Web+Book) | **88.54** | **78.23** | **71.82±2.58** | **76.77** | **67.32±1.21** | **70.09** | **52.01** |
>
> **Base Model (No SFT):** The Qwen3-8B-Base model performs substantially worse than the SFT-trained models across all benchmarks, providing a clear reference point for the observed gains.
>
> **Instruction-Tuned Model + continued SFT:** The results show an additional performance improvement, confirming that our high-quality synthetic data offers stronger reasoning capabilities even for already highly performant instruction-tuned models.

---

> ### Author Response · Authors · 2025-11-24
> **Response to Reviewer TpGs (5/5)**
>
> **For Question 3**
> >Question 3: Analysis of synthetic generated responses: Does the long-CoT Qwen3-235B responses ever disagree with the DeepSeek-R1 answer? If so, do you filter these examples out?
>
> For this question, we wish to present several considerations:
> 1.  A significant portion of the questions we synthesize are **complex reasoning problems** and **open-ended questions**, and our objective is to create a **multidisciplinary dataset**. Consequently, many disciplines, such as Humanities and Social Sciences, Applied and Professional Fields, and Arts, inherently involve open-ended questions that do not possess a fixed, single correct answer. This characteristic precludes a straightforward comparison of answer consistency. Therefore, we designed a prompt (detailed in the newly added Figure 16) and employed GPT-5.1 to evaluate the consistency between the reference answers generated by DeepSeek-R1-0528 and the long-CoT responses generated by Qwen3-235BA22B-Thinking-2507-FP8 for a random sample of 10,000 data points from our synthesized dataset. The consistency rate is 71.48%. We have incorporated this result into Section 4.4. Consequently, this result is reasonable given the inherent diversity of valid reasoning paths in open-ended multidisciplinary questions.
> 2.  Our combined datasets DLR-Book and DLR-Web contain approximately 4.7 million questions with very long long-CoT responses. Running comprehensive GPT-based analyses over the full datasets would be prohibitively expensive in both time and API cost.
> 3.  We would like to clarify that the reference answer is only a lightweight heuristic signal, not an authoritative ground truth. The concise reference answer generated by DeepSeek-R1-0528 is also self-generated by the model and is not guaranteed to be perfectly accurate. Moreover, the final precise answer may not explicitly reside within the original document, nor is it strictly required for the final answer to align with the original document to be correct. Therefore, a disagreement between the reference answer and the long CoT response does not definitively indicate which, if either, is erroneous. Our primary quality focus is ensuring that the question itself is answerable. We designed a prompt (detailed in the newly added Figure 14 of the paper) and used GPT-5.1 to check a random sample of 10,000 data points from our synthesized dataset. The proportion of questions deemed complete and answerable is **96.70%**. This shows that almost all questions are answerable.
> 4.  We also wish to clarify that the **response generation is not the primary focus of our paper**. Our main objective and proposed method concentrate on synthesizing **challenging and diverse multidisciplinary questions**. We believe that generating high-quality questions is often more critical than generating answers. Provided a high-quality question, the response can be generated using any model, including current state-of-the-art models such as GPT-5.1 and Gemini 3, or even more powerful future models, to attain higher response accuracy.
> 5.  Finally, we would like to underscore that our paper's main contribution lies in proposing the Design-Logic-Guided reasoning data synthesis pipeline for synthesizing challenging questions from raw corpora, synthesizing the corresponding large-scale dataset, and demonstrating its effectiveness. Our proposed methodology does not conflict with any quality control approach; subsequent strategies for quality control and data filtering can be applied to the existing synthesized data we provide. However, this is not the focus of our current work but is rather intended for future research. We included the reference answers to provide the community with the richest possible information, facilitating future research efforts to explore and utilize our dataset, such as for data selection studies.
>
> > Also, have you explored what happens if you include the corresponding design logics as problem-solving guidance (e.g. "for these types of problems some things to think about are A -> B -> C")
>
> Using design logic as an explicit problem-solving guide for model responses may be an interesting direction for future work. However, our current study focuses on demonstrating the effectiveness of "Design Logic" for question synthesis, rather than its contribution to model responses.
>
> ---
>
> >Small writing suggestion: The caption in Figure 6 ('source corpus quality') is ambiguous until you look at Section 5.5 (which discusses the acknowledged differences).
>
> Thank you for pointing out the ambiguity in the caption of Figure 6. We have revised the caption to be more explicit: "Figure 6: Performance comparison of models trained on data synthesized from lower-quality web corpora versus higher-quality book corpora."
>
> ---
>
> Finally, we would like to express our gratitude once again for the comprehensive suggestions you have provided. We have incorporated these suggestions in the revised paper. We hope our response addresses your concerns.

---

### Author Response · Authors · 2025-12-03
**Response Summary & Key Clarifications**

Dear ACs, SACs, and PCs,

We sincerely appreciate the reviewers' suggestions which have significantly improved our manuscript. We are pleased that two reviewers have already responded to our rebuttal, confirming that their concerns have been fully resolved and maintaining their positive rating (8 and 6). Although the remaining reviewers have not yet had the opportunity to reply, we are confident that our rebuttal has addressed their concerns as well.

We are grateful for the reviewers' recognition of **DESIGNER** as a novel and scalable framework for synthesizing challenging multidisciplinary reasoning data and for introducing reusable design logics that provide structured control over difficulty, diversity, and reasoning structure. We believe our work contributes a novel methodology and a high-quality multidisciplinary dataset.

We are encouraged by the reviewers' positive feedback, which highlights:

  * **Novelty & Motivation:** The "Design Logic" approach is novel, well-motivated, and provides a structured method for condensing multi-domain learnings (Reviewers TpGs, VsnW).
  * **Scalability & Utility:** The pipeline is systematic, scalable, and adaptable for other researchers (Reviewers TpGs, VTR2, mg22).
* **Effectiveness:** The resulting datasets are valuable, effectively creating challenging questions that significantly boost model performance (Reviewers TpGs, VsnW, wEhN, VTR2).
* **Insightful Analysis:** The detailed analyses on difficulty, diversity, and disciplinary coverage are informative and effective (Reviewers VsnW, wEhN).

---

To address the reviewers' main concerns and further strengthen our work, we have conducted additional experiments and clarifications, including:

**1. Comprehensive Data Quality Validation (Address Reviewers TpGs, mg22)**
To address concerns regarding data quality and label accuracy, we employed **GPT-5.1** to evaluate 10,000 randomly sampled instances (Section 4.4). The results confirm the high quality of our synthetic data:

  * **Question Answerability:** 96.70% of questions are complete and answerable.
  * **Design Logic Adherence:** 84.69% of questions strictly follow the target design logic.
  * **Label Accuracy:** High accuracy for Difficulty (96.53%), Question Type (91.22%), and Discipline (90.14%).

**2. Expanded Experiments (Address Reviewers TpGs, VTR2, wEhN)**

* **SFT baselines and base models.** We added results for base models (no SFT) and for continued SFT on official instruction-tuned models. Our data yields consistent gains regardless of whether SFT starts from base models or continues from instruction-tuned models. These results confirm that our data remains effective even for already strong models.
* **Full-Scale Ablation:** We added the "w/o Design Logic (Full)" experiment on the full 3.04M dataset. Results show a significant **5% performance gap** on GPQA-Diamond, proving the critical contribution of Design Logic.
* **Open-Ended Mathematical Reasoning Evaluation:** We added evaluations on GSM8K and MATH-500 (Appendix I). Although our dataset is built on raw corpora rather than expert-curated mathematical data and focuses on multidisciplinary coverage (mathematics constitutes only about 10 percent), it still achieves competitive performance on both MATH-500 and GSM8K.

**3. Reproducibility & Open Source (Address Reviewers VsnW, mg22):**

We reaffirm our commitment to open-sourcing the synthesized DLR dataset and the Design Logic Library. This allows the community to utilize our high-quality data directly or apply our method to new corpora. We have uploaded sample data and the full design logic library in the supplementary material.

**4. Clarification on Methodology (Address Reviewers TpGs, VsnW, mg22, VTR2)**

* **Core Scope:** We clarify that the core contribution of this paper is the **synthesis of challenging, diverse, multidisciplinary questions** guided by design logic, rather than response generation. Provided high-quality questions, the responses can be generated using any model, including more powerful future models, to attain higher response accuracy.
* **Generalizability:** Our filtering pipeline allows the method to work effectively on arbitrary question banks and public datasets, not just proprietary sources.
* **Compatibility with Quality Control:** Our work already includes standard quality-control mechanisms (labeling, deduplication, decontamination). Our methodology is fully compatible with any downstream quality control or data filtering strategies. While extensive quality control is not the primary focus of this work, we provide our large-scale datasets and a reusable design-logic library as foundational "seed" resources to facilitate future community research, such as data selection and training strategies.

---

We have integrated the responses into the revised paper, with changes highlighted in blue. We respectfully request that the ACs review the revised manuscript and provide a fair evaluation of our work.

---

### Meta-Review · Area_Chair_caGx · 2025-12-29

**Summary:**

This paper presents an ambitious and timely effort to construct a large-scale, multidisciplinary reasoning dataset using a structured “design logic” and an end-to-end LLM-driven generation pipeline. The core idea of explicitly guiding synthetic question generation with design logic is novel and conceptually well-motivated, and the resulting resource is substantially larger and more diverse than many existing reasoning benchmarks.

Despite some limitations—most notably the heavy reliance on prompt engineering and LLM self-judgment, as well as incomplete validation of difficulty labels and question quality—the empirical results indicate that the generated questions capture meaningful reasoning patterns and provide tangible downstream benefits. While evaluation comparability and noise control could be strengthened in future work, these issues do not fundamentally undermine the contribution.

Overall, the paper makes a significant and original contribution to the growing literature on synthetic reasoning data and LLM-based dataset construction.  I therefore recommend acceptance.

**Reviewer Concerns:**

The rebuttal partially addressed concerns regarding the overall motivation and design choices of the pipeline, clarifying the role of design logic in guiding question synthesis and providing additional qualitative justification for the usefulness of the generated questions. The authors also responded to questions about empirical gains by better contextualizing the downstream improvements and the intended use cases of the dataset.

However, several concerns remain outstanding. In particular, the heavy reliance on prompt engineering and LLM self-judgment is still not rigorously validated against human annotations or ground-truth metrics, and the reliability of difficulty labeling remains insufficiently substantiated. In addition, evaluation comparability across datasets and the accumulation of noise in the multi-stage construction pipeline are not fully resolved, though these limitations do not negate the overall contribution.

**Reviewer Scores:**

Overall, I expect most reviewers would have maintained their original scores or increased them slightly (e.g., by +2) after discussion, given the clarified motivation and acknowledged but non-fatal limitations.
The scores may be 4,4,6,6,8

---

### Decision · Program_Chairs · 2026-01-26

Accept (Poster)